SOFTWARE

# MCell4 with BioNetGen: A Monte Carlo simulator of rule-based reaction-diffusion systems with Python interface

Adam Husar[1], Mariam Ordyan[2], Guadalupe C. Garcia[1], Joel G. Yancey[1], Ali S. Saglam[3], James R. Faeder[3], Thomas M. Bartol[1]\*, Mary B. Kennedy[4], Terrence J. Sejnowski[1,2]

1 Computational Neurobiology Lab, Salk Institute for Biological Studies, La Jolla, California, United States of America, 2 Institute for Neural Computations, University of California, San Diego, La Jolla, California, United States of America, 3 Department of Computational and Systems Biology, University of Pittsburgh, Pittsburgh, Pennsylvania, United States of America, 4 Division of Biology and Biological Engineering, California Institute of Technology, Pasadena, California, United States of America

\* bartol@salk.edu

**Data Availability Statement:** Easy installation package for all major computing platforms are available at mcell.org/download.html Source code is available at github.com/mcellteam/mcell github.

## Abstract

Biochemical signaling pathways in living cells are often highly organized into spatially segregated volumes, membranes, scaffolds, subcellular compartments, and organelles comprising small numbers of interacting molecules. At this level of granularity stochastic behavior dominates, well-mixed continuum approximations based on concentrations break down and a particle-based approach is more accurate and more efficient. We describe and validate a new version of the open-source MCell simulation program (MCell4), which supports generalized 3D Monte Carlo modeling of diffusion and chemical reaction of discrete molecules and macromolecular complexes in solution, on surfaces representing membranes, and combinations thereof. The main improvements in MCell4 compared to the previous versions, MCell3 and MCell3-R, include a Python interface and native BioNetGen reaction language (BNGL) support. MCell4's Python interface opens up completely new possibilities for interfacing with external simulators to allow creation of sophisticated event-driven multiscale/multiphysics simulations. The native BNGL support, implemented through a new open-source library libBNG (also introduced in this paper), provides the capability to run a given BNGL model spatially resolved in MCell4 and, with appropriate simplifying assumptions, also in the BioNetGen simulation environment, greatly accelerating and simplifying model validation and comparison.

This is a *PLOS Computational Biology* Software paper.

## 1 Introduction

Living cells are complex structures in which biomolecules and biochemical processes are spatially organized and span the extracellular space, plasma membrane, cytosol, and subcellular

com/mcellteam/cellblender Source code and data for examples presented in the manuscript are available at github.com/mcellteam/article_mcell4_1.

**Funding:** Funding for this research was provided by NIH MMBioS P41-GM103712 (TJS, TMB, AH, GCG, MO, JGY, JRF, AS), NIH CRCNS R01-MH115556 (TJS, TMB, MO, MBK), NIH CRCNS R01-MH129066 (TJS, TMB, MO, MBK), NSF NeuroNex DBI-1707356 (TJS, TMB, AH, GCG, MO, JGY), and NSF NeuroNex DBI-2014862 (TJS, TMB, AH, GCG, MO, JGY). The funders had no role in study design, data collection and analysis, decision to publish, or preparation of the manuscript.

**Competing interests:** The authors have declared that no competing interests exist.

organelles. These biochemical processes are intrinsically multiscale in nature because they are based on molecular interactions on a small scale leading to emergent behavior of cells on a larger scale. Because of the dynamic nature of biochemical processes on different temporal and spatial scales, appropriate mathematical tools are required to understand the underlying dynamics and to dissect the mechanisms that control system behavior [1]. Overall, understanding how cellular design dictates function is essential to understanding health and disease in the brain, heart, and elsewhere. MCell (Monte Carlo Cell) is a biochemistry simulation tool that uses spatially realistic 3D cellular models and stochastic Monte Carlo algorithms simulation of the movements and interactions of discrete molecules within and between cells [2–5]. Here we describe MCell4, a new version of MCell.

One of the most important new features in MCell4 is a flexible Python application programming interface (API) that allows coupling between MCell and other simulation engines or other custom code. By itself MCell performs particle-based reaction-diffusion simulations on spatial and temporal scales from nm to μm and from μs to 10s of seconds. MCell4's Python API extends its capabilities and provides means to define multiscale hybrid models that can allow to simulate larger systems on longer timescales, as we demonstrate here with a simple example.

A second important addition to MCell4 is efficient support for rule-based modeling by making use of the BioNetGen (BNG) Language (BNGL). BNG is an open-source software package for representing and simulating biochemical reactions [6]. Although powerful, BNG models are non-spatial. Support for models implemented in BNGL within MCell4 permits determination of the role of space in different reaction scenarios. This is not a trivial task because the time scales of diffusion and of reactions [7], as well as the spatial localization of proteins, influence the results.

We first present the design principles of MCell4 and its API, and next we introduce a new BioNetGen library. Finally, we demonstrate some of the new features in MCell4 with examples and present a simple hybrid model that couples spatial simulations in MCell4 with ordinary differential equations (ODEs).

## 1.1 Particle-based reaction dynamics tools

In particle-based reaction-diffusion simulations, each molecule is represented as an individual, spatially resolved, autonomous agent. In this sense, particle-based simulations are a subset of agent-based simulations. Molecules diffuse either within volumes or on membrane surfaces and may affect each other by reacting upon collision. A review of currently maintained particle-based stochastic simulators which describes Smoldyn [7], eGFRD [8], SpringSaLaD [9], ReaDDy [10], and MCell3 was recently published in [11]. It should be noted that some of the new features of MCell4 that we report here are available in the current version of Smoldyn, v2.72. In particular, Smoldyn has a Python API with callback functions, allows transmembrane bimolecular interactions, and supports BNGL via pre-generation of the full reaction network. In contrast, MCell4 integrates BNGL via a Network-Free method with on-the-fly direct reaction rule evaluation [12], resulting in dramatic performance improvements.

MCell is a particle-based simulator that represents volume molecules as point particles and surface molecules as area-filling tiles on surfaces. The typical simulation time-step in MCell is 1 μs, and the simulated times can stretch from milliseconds to minutes. The Monte Carlo reaction-diffusion algorithms employed in MCell were first introduced in [13] and discussed further in [3, 4, 14].

Briefly, MCell operates as follows. MCell tracks the unimolecular state transitions of individual molecules as well as the bimolecular collisions and interactions of individual diffusing

molecules. The expected rate of encounter between individual colliding molecules can be calculated from the solution to the diffusion equation in unbounded space (see [3]). The calculated rate of encounter and the user-specified mass-action kinetic bimolecular reaction rate constant for a given reaction are then used to calculate the probability of reaction per collision event. As a volume molecule diffuses through space by random Brownian motion, all volume molecules within a given radius (i.e. the interaction radius, $r_{int}$) along its trajectory, or the single surface molecule located at the point of collision on a surface, are considered as possible reaction partners. As a surface molecule diffuses it is first moved to its final position after one time step and any surface molecules immediately adjacent to that final position are considered as possible reaction partners. Molecules diffusing in 3D volumes do not themselves have volume (i.e. no volume exclusion). The collision cross-section area for interactions among volume molecules is derived from $r_{int}$. Molecules on membranes (i.e. surfaces) occupy a fixed area defined by the individual triangular grid elements (tiles) created by subdividing the surface mesh triangles with a barycentric grid. The collision cross-section for interactions between volume and surface molecules and among surface molecules is derived from the density of the barycentric surface grid. At runtime MCell reports the reaction probabilities for bimolecular reactions and warns if probabilities are greater than 1. And at the end of the simulation MCell reports the estimated fraction of missed reactions caused by high probabilities. The probability of reaction per collision for a given reaction depends on the diffusion constant and reaction rate constant, which are properties of the system being modeled, and the time step, which is chosen by the researcher. We have found that for most bimolecular interactions, a time step of 1 μs is appropriate.

MCell is able to represent arbitrary geometries comprised of triangulated surface meshes. Thus complex models such as a 180 μm$^3$ 3 dimensional serial electron microscopic reconstruction of hippocampal neuropil have been used to construct a geometrically-precise and biophysically accurate simulation of synaptic transmission and calcium dynamics in neuronal synapses [5]. A detailed description of the mathematical foundations of MCell's algorithms can be found in these references [2–4, 14].

MCell3-R [15], a precursor of MCell4, is an extension of MCell that supports BNGL [16] and allows modeling of protein complexes or polymers by using rule-based definition of reactions. MCell3-R uses a library called NFsim [12] to compute the products of reaction rules for reactions described in BNGL.

MCell4 is an entirely new implementation of MCell, written in C++ (using the C++17 standard). It provides a versatile Python interface in addition to many other improvements. In particular it runs dramatically faster when simulating complex reaction networks expressed as rules in BNGL. And all of the features of MCell that were introduced previously [4] have been retained, either directly in the implementation or indirectly via callback functions (discussed in more detail below, see also Table A in Supporting Information S1 Text). Here we briefly describe the motivations for introducing new features in MCell4.

## 1.2 Motivation for the MCell4 Python application programming interface

We had two important motivations for the creation of the MCell4 Python API: 1) to give the users the freedom to customize their models in a full-featured modern programming language, and 2) to create an easy way to couple MCell4 with other simulation platforms to allow multi-scale, multi-physics simulations.

The main goal when designing the new API for MCell4 was to allow definition of complex models combining many reaction pathways distributed over complex geometry. Thus, a main requirement was to enable modularity with reusable components that can be independently

validated. With this feature, one can build complex models by combining existing modules with new ones.

As in the approach in the PySB modeling framework [17], a model in MCell4 is seen as a piece of software, allowing the same processes used in software development to be applied to biological model development. The most important such processes are: 1) incremental development where the model is built step by step, relying on solid foundations of modeling that has been validated previously, 2) modularity that provides the capability to create self-contained, reusable libraries, 3) unit testing and validation to verify that parts of the model behave as expected, and 4) human-readable and writable model code that can be stored with git or other code version control software. In addition to being essential for incremental development, this also allows code reviews [18] so that other team members can inspect the latest changes to the model and can contribute their own modules to the growing code base.

## 1.3 Motivation for a new BioNetGen library

NFsim [12] is a C++ library that provides BioNetGen support, implements the network-free method, and is used in MCell3-R [15]. To use a BNGL model in MCell3-R, the BNGL file first needs to be parsed by the BioNetGen compiler; then, a converter generates a file containing MCell Model Description Language (MDL), a file with rule-based extensions to MDL (MDLR), and additional XML files required by the NFsim library. These files then constitute the model for simulation in MCell3-R. The disadvantage of this approach is that the original BNGL file is no longer present in the MCell3-R model. Thus each time changes are made to the model, the external converter must be run again, and any changes made by hand to the MCell3-R model files will be lost. Without an API for interacting with BNG directly, MCell3-R has no means to overcome this issue. MCell3-R also has performance and memory consumption problems when the simulated system has a large number of potential reactions.

To create a seamless integration of BNGL with MCell4 we implemented a new library for the BioNetGen language that contains a BNGL parser and a network-free BNG reaction engine the main purpose of which is to compute reaction products for a given set of reaction rules and reactants. This BNG library (libBNG) is implemented in C++17 and was designed to be independent of MCell4 in mind so that it can be used in other simulation tools. libBNG does not yet support all of the special features and keywords of the BioNetGen tool suite (see also Table B in Supporting Information S1 Text). Most notably, BNGL functions are not supported, however the set of supported features is sufficient for any MCell4 model. And when a special function is needed, it can be represented in Python code with the MCell4 API, which has access to the full BNGL representation of any given model. The source code of libBNG is available under the MIT license in Reference [19].

## 1.4 Features of MCell4

Here we briefly describe some of the features of MCell4. In the results section we present a few relevant examples specifically to demonstrate and validate the accuracy of some of these features. In this section we indicate which example illustrates the mentioned feature. For those new features not specifically demonstrated due to space limitations in this article, we provide links to end-user documentation, tutorials, and source code repositories containing detailed information and validation tests.

**1.4.1 Python/C++ API for model creation and execution.** All models can now be created in Python. CellBlender (see section 1.5) is useful for creation of many relatively simple models without the need to write Python code by hand. However, more complicated customized models will need to include a custom Python script. While CellBlender provides for inclusion of

custom Python scripts, for simplicity and explanatory power, all the examples presented in the results section of this paper are written solely in Python.

**1.4.2 Reactions are now written in BNGL.** In MCell4 the reaction language is BNGL [16]. Thus, MCell4 fully supports rule-based reactions and all models use this feature.

Most importantly, the support for BNGL and NFsim means that MCell4 performs direct, agent-based evaluation of reaction rules and thus enables spatially-resolved network-free simulations of interactions between and among volume and surface molecules. The CaMKII holoenzyme model in the results section 3.1.3, for example, would not be possible without the spatial network-free algorithms implemented in MCell4.

**1.4.3 Ability to go back and forth between MCell4 and BNG simulator environments.** The new BNG library [19] allows direct loading and parsing of a BNG model that can then be placed within a realistic 3D cellular geometry. This allows comparison of the results of non-spatial (simulated with BioNetGen solvers) and spatial (simulated with MCell4) implementations of the same BNG model. Two of the examples in the results section demonstrate this feature: SNARE (3.1.1), and CaMKII (3.1.3).

If the spatial features are found to be unimportant for a given model, and simulation speed is of more concern, the BNGL file can be run as a separate module with the BNG simulator. See section 2.4.4 for an example.

**1.4.4 Other advanced features.** Among the more advanced features introduced in MCell4 is the ability to implement transcellular and transmembrane interactions that occur between surface molecules located on separate membranes (for details see documentation at mcell.org [20], and the example in the test suite [21]). MCell4 also supports both coarse-grained and fine-grained customization of models by customizing the time-step and by introducing event-driven callbacks. Callbacks implement custom Python code that runs when a particular reaction occurs or when a collision occurs between a molecule and a surface. An example of the use of callbacks to implement release of neurotransmitters when a SNARE complex is activated is shown in section-3.1.2.

Finally, the new Python API supports the ability to create multi-scale multi-physics hybrid simulations that take advantage of all the existing Python packages. For an example of a simple hybrid model see section 3.3.

## 1.5 Model creation and visualization in CellBlender

CellBlender is a Blender [22] addon that supports creation, execution, analysis, and visualization of MCell3/MCell3-R and MCell4 models. CellBlender still supports creation and execution of MCell3/MCell3-R models and has been updated to include several new features for MCell4: automatic generation of well-structured Python and BNGL code from the CellBlender representations of complete MCell4 models; execution, analysis, and visualization of these models; and visualization of simulation data generated by simulations of externally created Python-only models. CellBlender offers an easy way to begin using MCell through built-in examples (Fig 1 shows an example of a model of the Rat Neuromuscular Junction included in CellBlender), and tutorials [23].

## 2 Design and implementation

### 2.1 MCell4: A bird's eye view

We will briefly review MCell4's architecture and fundamental aspects of its API, starting with Fig 2.

MCell simulations progress through time by a series of iterations. The duration of an iteration is given by a user-defined time step (usually 1 μs). The Scheduler keeps track of events to

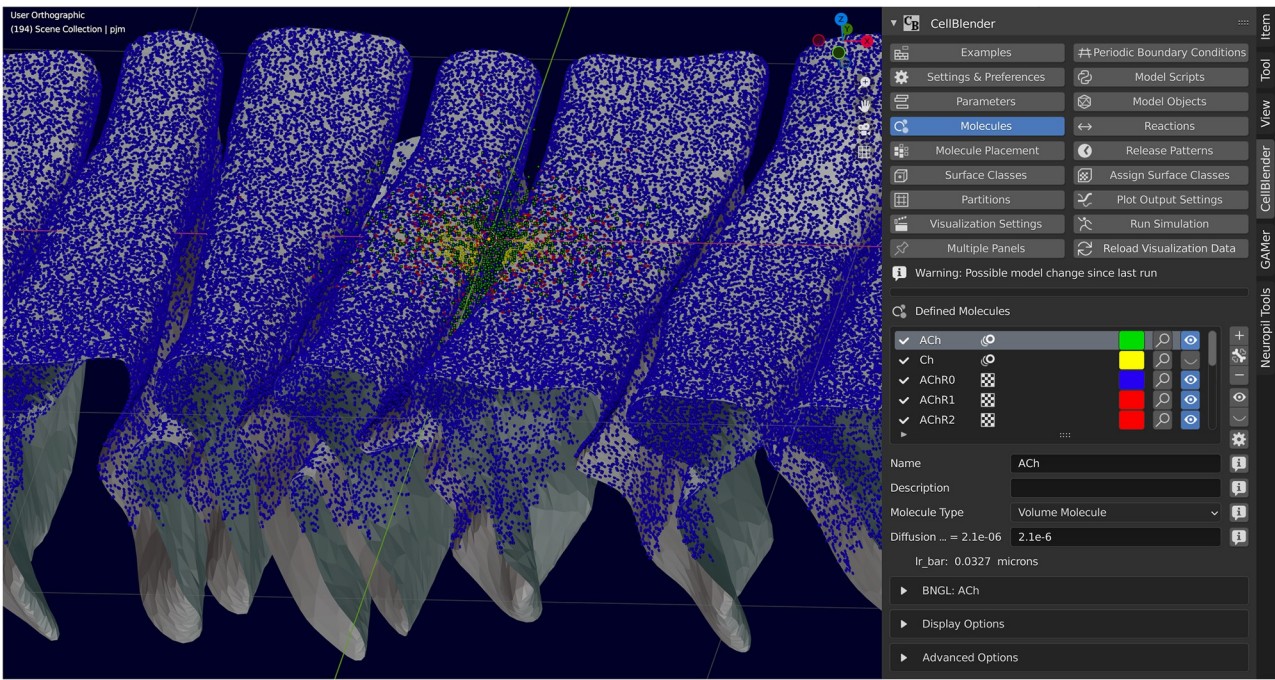

**Fig 1. Model creation with CellBlender.** MCell4 models can be created, executed, analyzed, and visualized using CellBlender, an addon for Blender. The capabilities of Blender are indispensable for creating complex geometries for MCell4 models.

be run in each iteration. The main simulation loop implemented in the all-inclusive object called "World" requests the Scheduler to handle all the events that occur in each iteration (Fig 3) until the desired number of iterations has been completed.

## 2.2 Python API generator: A closer look

The MCell4 physics engine is implemented in C++. To ensure reliable correspondence between the representation of a model in Python and in C++, we have implemented a Python

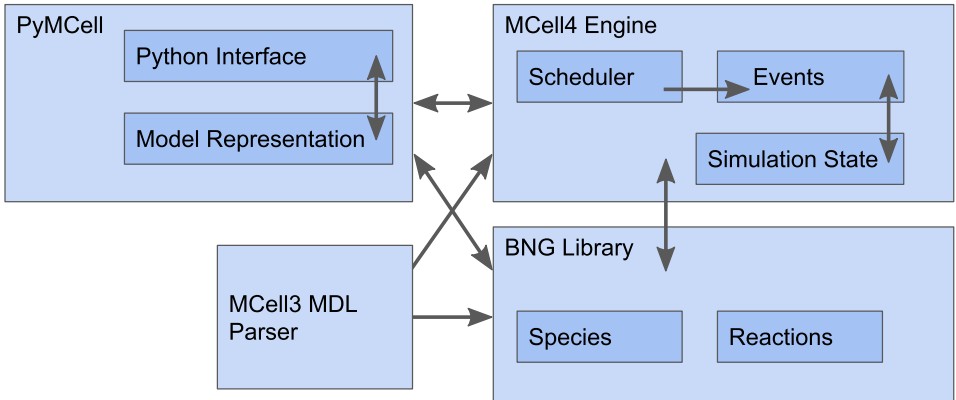

**Fig 2. Overview of MCell4 architecture.** MCell4 is comprised of four main components: 1) The PyMCell library provides a Python interface and contains classes to hold the model representation, 2) The MCell4 engine implements the simulation algorithms, 3) The BNG (BioNetGen) library provides methods to resolve BioNetGen reactions, and 4) The MDL (Model Description Language) parser enables backward compatibility with MCell3.

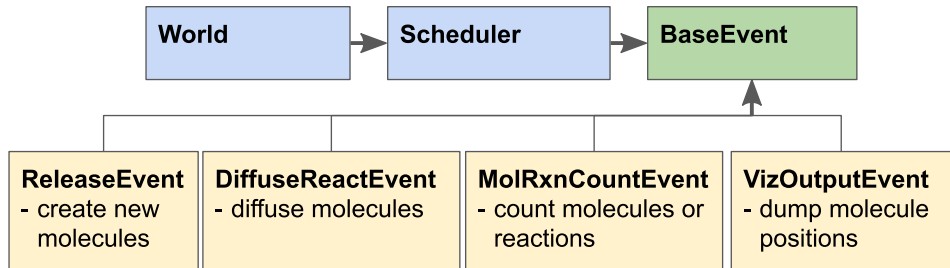

**Fig 3. Overview of MCell4 Scheduler.** The Scheduler executes time step iterations which consist of discrete events executed in this order: 1) A ReleaseEvent creates new molecules, 2) A DiffuseReactEvent implements diffusion of molecules, checks collisions, and executes reactions, 3) A MolRxnCountEvent counts numbers of molecules or how many times a reaction occurs, and 4) A VizOutputEvent stores molecule locations for visualization in CellBlender. Only the DiffuseReactEvent must be executed at each time step to move the time forward. The other events listed here are optional.

API generator which is used when building the MCell4 executable and Python module from source code. The API generator reads a high-level definition file in the YAML format and automatically generates all the base C++ classes, their *corresponding* Python API representations for the pybind11 library [24], code for informative error messages, and end-user documentation. A consistent Python and C++ API contributes to the quality of the user experience when creating a model, and facilitates well-maintained end-user documentation.

The presence of the API generator, schematically represented in Fig 4, ensures that when new features are added to MCell4, one only needs to modify a single API definition in the YAML format to ensure that both the API and the end-user documentation reflect the new features.

## 2.3 MCell4 model structure

A predefined model structure is important to enable reusability of model components (e.g., [28]). With a predefined model structure every piece of code for a given component (such as

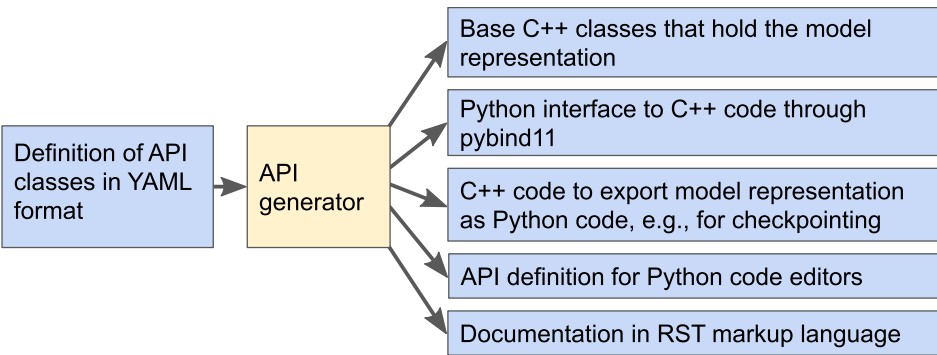

**Fig 4. Overview of MCell4 API generator.** When MCell4 is built from its source code, the API generator reads a high-level definition of the MCell4 Python interface and generates code and documentation. Automatic generation of an API makes it possible to easily modify or extend the API while ensuring that all parts including documentation stay consistent. While the generated C++ and pybind11 code are not directly relevant to users, the generated API definition for Python code editors (contained in the file mcell.pyi) enables code completion and parameter info features in Python editors that support it such as VSCode [25]. This is indispensable to users creating their own models using the MCell4 Python API. Furthermore, the generated end-user documentation provides necessary details on what classes and methods are offered [26]. Checkpointing stores the current state of simulation as an MCell4 model in Python that can be manually modified if needed. The API generator is a general tool that can also be used (with minor modifications) for other software tools that combine C++ and Python [27].

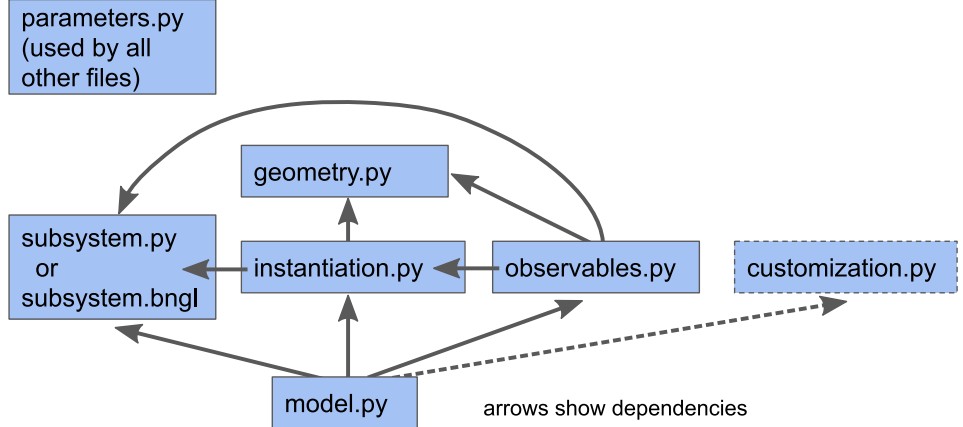

**Fig 5. Recommended structure of a standard MCell4 model.** The main files included in a standard MCell4 model are: 1) parameters.py with all the model parameters, 2) subsystem.py that captures information on species and reactions in a way that is independent of a particular model and can be used as a reusable module, 3) geometry.py with a definition of 3D geometry objects, 4) instantiation.py that usually defines the initial model state, i.e., which geometry objects are created in the simulation and the number and positions of molecules to be released at a given time, 5) observables.py with lists of characteristics to be measured and saved in files during simulation, and 6) model.py in which all the parts of the model are assembled together and in which the the simulation loop with optional interactions with external simulators is defined. This recommended model structure is not enforced by MCell4 itself. However, use of this structure promotes readability, modularity, navigation, and sharing of models with others. The MCell4 Python code generator in CellBlender follows this recommended structure with the addition of a file called customization.py whose purpose is to provide a means to modify/expand the model in the case when the standard files are automatically generated from the CellBlender representation.

reaction definitions, geometry, initial model state, and observables) is in a file with a specified name and follows a predefined coding style. Such standardized model structure (shown in Fig 5) aids in the reuse of code and simplifies creation of new models by leveraging existing model components. Another advantage of a predefined model structure is the capability to combine parts of existing models into one model (Fig 6).

**2.3.1 Example model using the MCell4 Python API.** A simple example that shows the MCell4 API including Subsystem, Instantiation, and Model classes is shown in Fig 7. Because of the simplicity of this example, we do not show the division into the separate files illustrated in Fig 5.

## 2.4 Graph-based approach to protein modeling

BNGL [30] supports intuitive modeling of protein complexes by representing them as undirected graphs. Such graphs contain two types of nodes: *elementary molecules* and *components*. Component nodes represent *binding sites* of the protein and can also express the *state* of the whole protein or of a binding site. A graph representing a *single protein* is implemented as an elementary molecule node with component nodes connected to it through *edges*. To form a dimer, two individual components of different proteins are bound by creating an edge between them. A graph with one or more elementary molecules with their components is called a *complex*. A *reaction rule* defines a graph transformation that manipulates the graph of reactants. A reaction rule usually manipulates edges to connect or disconnect complexes or change the state of a component. It can also modify the elementary molecules such as in the reaction A + B -> C where we do not care about the molecular details and do not need to model individual binding sites. An example of applying a reaction rule that connects complexes and changes the state is shown in Fig 8. Note that what we call an "elementary molecule type" here is called

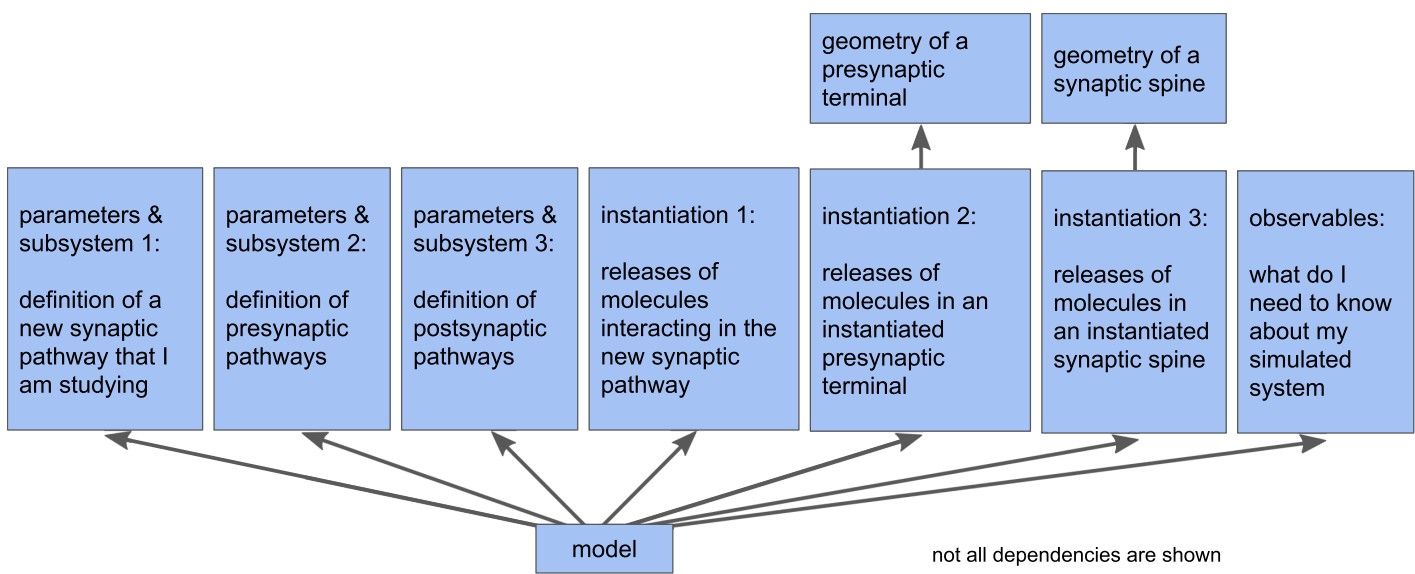

**Fig 6. Example of model modularity.** Modularity of a model allows assembly of multiple subsystem definitions into a single model. In the example shown here, individual modules are assembled to construct a model of a new synaptic pathway that is affected by other processes. The complete model includes modules that individually define the presynaptic terminal with its presynaptic pathways and the postsynaptic spine with its postsynaptic pathways.

a "molecule type" in BioNetGen. In MCell, "molecules" are defined as whole molecules such as protein complexes that act as individual agents in the simulation. For better clarity, we adopt the name "elementary molecule" for the base building blocks of complexes. The tool SpringSa-LaD [9] uses the same distinction.

This graph-based approach is essential when dealing with combinatorial complexity. To model a protein that has 10 sites, in which each can be unphosphorylated, phosphorylated, or bound to another protein with ordinary differential equations (ODEs) requires $3^{10}$ (i.e. 59049) simultaneous ODEs [31]. For comparison, a BNGL model of the same protein will have just 6 reversible reaction rules (assuming no interaction between these 10 sites). Such a model can then be simulated using network-free simulation methods [32]. MCell4 uses on-the-fly network expansion (i.e. the NFsim direct rule evaluation method [12]) and creates complex species and applicable reactions only when they are needed. Fig 8 describes how libBNG resolves and expands the individual reaction rule corresponding to a pair of colliding molecular species directly in just-in-time fashion. The results of rule expansion are cached for efficiency. But to free-up memory, the cached list of species and applicable reactions is periodically cleaned-up when no molecules of such species exist in the simulation. It is critical to note that the whole reaction network is never expanded as this would fail in the case of pathological combinatorial explosion, for example with linear or branched polymers, or in case of complex molecules like CaMKII (see section 3.1.3).

**2.4.1 Extension of BNGL for volume-surface reactions.** BNGL compartments [33] allow the definition of hierarchical volumes or surfaces where simulated molecules are located. To model a transport reaction that moves a volume molecule from one compartment through a channel (located in a membrane) into another volume compartment, one must specify the direction of this transport. We show such a reaction which implements hierarchy of compartments in Fig 9.

In BNGL, a reaction that defines the transport of A from compartment EC into CP through transporter T is represented with the following rule:

**MCell4 Python API**

```python
import mcell as m

subsystem = m.Subsystem()
a = m.Species(
    name = 'a',                     # this species will be called 'a',
    diffusion_constant_3d = 1e-6    # molecules of 'a' are volume
                                    # molecules and diffuse in 3D space
)
subsystem.add_species(a)

instantiation = m.Instantiation()
# ReleaseSite defines which and how many molecules will be released
# either when simulation starts (default) or at a predefined time
rel_a = m.ReleaseSite(
    name = 'rel_a',
    complex = a,              # molecules of which species to release
    number_to_release = 10,  # copy number
    location = (0, 0, 0)      # all these molecules will be released
                             # at (x, y, z) location (0, 0, 0)
)
instantiation.add_release_site(rel_a)

model = m.Model()
model.add_subsystem(subsystem)          # include information on species
model.add_instantiation(instantiation)  # include molecule release site

model.initialize()                      # initialize simulation state
model.run_iterations(10)                # simulate 10 iterations
model.end_simulation()                  # final simulation step
```

**Fig 7. A simple MCell4 model.** Example of a simple MCell4 model that releases 10 volume molecules of species 'a' and simulates their diffusion for 10 iterations with a default time step of 1 μs. Note that for this and the following examples, a system variable, PYTHONPATH, must be set so that the Python interpreter knows where to find the MCell4 module. Further details are provided in the MCell4 Installation Documentation [29]. MCell4/CellBlender is provided as a bundle which includes Blender, CellBlender, and MCell4 all packaged together. Blender comes with its own Python built-in. MCell4 can use this Python.

```
A@EC + T@PM -> A@CP + T@PM
```

To model multiple instances of cells or organelles, this definition needs to be replicated with different compartments as follows:

```
A@EC + T@PM1 -> A@CP1 + T@PM1
A@EC + T@PM2 -> A@CP2 + T@PM2
...
```

MCell3 uses a general specification of orientations [4] in which the rule above is represented as:

| (A) Reaction rule | $A(\mathbf{c0{\sim}R})$ + $B(\mathbf{c1})$ -> $A(\mathbf{c0{\sim}S!1}).B(\mathbf{c1!1})$ |
|---|---|
| (B) Reactants and product | $A(c0{\sim}S,\mathbf{c0{\sim}R})$ + $B(\mathbf{c1{\sim}U},c2{\sim}X)$ -> $A(c0{\sim}S,\mathbf{c0{\sim}S!1}).B(\mathbf{c1{\sim}U!1},c2{\sim}X)$ |

(C) Graph representation of reactants and products

(D) Reactants

(E) Rule - reactant patterns

$A(\mathbf{c0{\sim}R})$ + B$(\mathbf{c1})$

1) Map reactant patterns onto reactants

(F) Rule - product

$A(\mathbf{c0{\sim}S!1}).B(\mathbf{c1!1})$

2) Map product(s) onto reactant patterns and determine changes

(G) Reactants changed into products

3) Apply changes determined in step 2) onto reactants

**Fig 8. Example of graph transformation with BNG reaction rules.** This example shows details of how BNG reactions are implemented in the libBNG library [19]. Reactants are defined with molecule types $A(c0 \sim R \sim S, c0 \sim R \sim S)$ and $B(c1 \sim U \sim V, c2 \sim X \sim Y)$ where A and B are names of the molecule types, c0 is a component of A that can be in one of the states R and S, and similarly c2 and c3 are components of B. (A) is the example reaction rule, (B) are example species reactants and products in the BNGL syntax, and (C) shows a graph representation of the rule in (B). Application of the rule is done in the following steps: 1) a mapping from each molecule and each component from reactant patterns (E) onto reactants (D) is computed (dotted arrows). If the state of a component is set in the pattern, the corresponding reactant's component state must match. The next step 2) is to compute a mapping of the rule product pattern (F) onto reactant patterns (E). The difference between the reaction rule product pattern and the reactant patterns tells what changes need to be made to generate the product. In this example, a bond between A's component c0 with state R and B's component c1 is created. The state of A's component c0 is changed to S. Once the mappings are computed, we follow the arrows leading from the reaction rule product pattern (F) to reactant patterns (E) and then to reactants (D) and 3) perform changes on the reactants resulting in the product graph (G). Each graph component of the product graph is a separate product and there is exactly one product in this example.

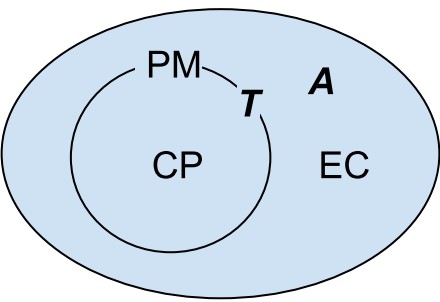

**Fig 9. Example of compartments.** EC is extracellular space, PM is the plasma membrane, and CP is cytoplasm. A is a molecule that diffuses freely in 3D space, and T is a molecule located in the plasma membrane.

```
A' + T' -> A, + T'
```

On the reactant side of the reaction, A' (A followed by an apostrophe) means that molecule A hits molecule T from the "outside" (as defined below) of the compartment, and T' means that the surface molecule T must be oriented in the membrane facing towards the outside. On the product side of the reaction, A, (A followed by a comma) means that the product A will be created on the inside of the compartment and T' means that T will still be oriented towards the outside. Geometric objects in MCell are composed of triangles. The "outside" of a triangle is defined as the direction in which the normal vector of the triangle points. More details on molecule orientations defined in MCell3 can be found in [4].

Because the MCell3 representation of orientation is not compatible with the grammar of BNGL, and to avoid repetition of reaction rules for each compartment, we have defined an extension to BNGL that allows two special compartment classes called @IN and @OUT to be used in MCell4. With this extension reactions with compartments are then more generally defined as:

```
A@OUT + T -> A@IN + T
```

Note that only bimolecular reactions where a volume and a surface reactant interact may use the @IN or @OUT compartment classes. When the rule is used at simulation time the actual membrane compartment containing the surface reactant (T here) along with the volumetric compartment containing the volume reactant (A here) are used to correctly interpret the geometric meaning of the @IN and @OUT compartment class associated with the volume reactant. For example when this rule is applied to reactants A@EC (i.e A located in EC) and T@PM (i.e. T located in PM) at simulation time, MCell4 will first interpret the @OUT compartment class in the rule and find that compartment EC is outside of PM, and satisfies the left-hand side of the rule. Next MCell4 finds that compartment CP is inside of PM, and finalizes the mapping of the generic compartment class @IN to the specific compartment class @CP MCell4 then inserts this specific compartment information into the rule A@OUT + T -> A@IN + T to get the runtime rule A@EC + T@PM -> A@CP + T@PM which is the same as the example rule we started with.

One more situation that we considered is how to define the orientation of the transporter in the membrane. One might need to model flippases and floppases (e.g., [34]) that change the orientation of a receptor in a membrane. In MCell4, when a molecule is created in a membrane, its orientation is always facing outwards (equivalent to T' in the MCell3 notation). If one needs to define orientation explicitly, a component of an elementary molecule can be defined. For example one can extend the definition of the molecule type T to contain a component called 'o' with two states called INWARDS and OUTWARDS. The rule defined for a specific state of the transporter will then be:

```
A@OUT + T(o~OUTWARDS) -> A@IN + T(o~OUTWARDS)
```

To flip the orientation of T, a standard BNGL rule F + T(o $\sim$ OUTWARDS) -> F + T (o $\sim$ INWARDS) can be defined; Here, F is a surface molecule flippase.

To summarize, we introduced an extension to BNGL in which compartment classes @IN and @OUT are used to define general volume+surface molecule reaction rules that can be applied to any specific compartments at simulation time.

The original MCell3 notation with commas and apostrophes is not supported in BNGL. However, rules that use this notation are allowed in CellBlender and CellBlender will export these rules directly into their corresponding Python representation.

**2.4.2 Units and interoperability between MCell4 and BioNetGen.** Usage of the BioNetGen language offers an excellent interchange format. Model definitions in BNGL can be executed by MCell, and BioNetGen itself implements various simulation approaches such as ODE, SSA [35], PLA [36], and NFsim [12]. BioNetGen does not have a predefined unit system

so that the user is free to use any unit system they deem suitable and that is compatible with the underlying algorithms. Indeed, some of the solvers employed by BNG require special attention regarding units. For example NFsim requires bimolecular reaction rate constants in units of $N^{-1}s^{-1}$ (where N is particle number), and the ODE solver and SSA solvers are most stable when supplied with units of $\mu m^3\,N^{-1}s^{-1}$. To facilitate model interchange, we define a set of units to be used when BNGL models are implemented in MCell4 and when the model is exported for use within BioNetGen as shown in Table 1.

An MCell4 model is typically implemented as a combination of Python and BNGL code. The recommended approach is to define all the molecule species and reaction rules by BNGL, but the MCell4 Python API provides means to define them programmatically as well. There are also aspects of spatial models that cannot be captured by BNGL. To simplify model validation, MCell4 provides an automated means to export a model that has been implemented as a combination of Python and BNGL into pure BNGL. Since not all features (especially spatial distributions) of an MCell4 model can be mapped to pure BNGL, a best-effort approach is used during this export. All model features that can be translated into BNGL are exported and error messages are printed identifying the model aspects that have no equivalent in BNGL. In general, no model component requiring specific location in 3D space can be exported to BNGL. If there are no such components and the model was exported without errors, the exported BNGL can be used for cross-validation and comparison of results between the MCell4 and BioNetGen simulations. Verifying results with multiple tools can reveal errors in the model or in the simulation tools. Therefore, such validation is a recommended step in development of an MCell4 model.

**2.4.3 Determination of diffusion constants for BNGL complexes.** Elementary molecules are assigned a diffusion constant at the time of their definition in a model. As these molecules bind to form complexes, MCell4 derives the diffusion constant of the newly formed complex from its elementary constituents. The computation of diffusion constants of complexes is done according to combining rules and equations published in [15]. If a complex contains any species that has a diffusion constant equal to zero then the entire complex will have a diffusion constant of zero. If a complex contains any surface species, then the whole complex is considered to be a surface complex. In this case only the surface subunits are used to derive the surface diffusion constant. If there are no surface subunits, the complex is considered to be a volume complex and all the volume subunits partake in computation of the volume diffusion constant.

**2.4.4 Example of an MCell4 model with BioNetGen specification.** To demonstrate the support for BNGL in MCell4, we show a simple BNGL file (Fig 10) whose definitions of species and reaction rules, molecule releases, and compartments are imported (i.e. loaded) into an MCell4 model as shown in (Fig 11).

**Table 1. Units used in MCell4 and suggested units for BioNetGen.** Unit N represents the number of molecules and M is molar concentration. BioNetGen interprets membranes (2D compartments) as thin volumes of thickness 10 nm. NFsim in BioNetGen does not fully support compartmental BNGL yet and the volume of the compartment must be incorporated into the rate units of the reactions occurring in that compartment, therefore NFsim's bimolecular reaction rate unit does not contain a volumetric component. Additional units in MCell4 include: length in μm and diffusion constants in $cm^2\,s^{-1}$.

| Simulation tool and mode of usage | Volume-volume or volume-surface bimolecular reaction rate | Surface-surface bimolecular reaction rate | Unimolecular reaction rate | Compartment volume | Seed species (initial molecule release) value |
|---|---|---|---|---|---|
| MCell4 with default units | $M^{-1}\,s^{-1}$ | $\mu m^2\,N^{-1}s^{-1}$ | $s^{-1}$ | $\mu m^3$ | N |
| MCell4 with BNG units; BioNetGen ODE, SSA, PLA | $\mu m^3\,N^{-1}s^{-1}$ | $\mu m^3\,N^{-1}s^{-1}$ | $s^{-1}$ | $\mu m^3$ | N |
| BioNetGen NFsim | $N^{-1}s^{-1}$ | $N^{-1}s^{-1}$ | $s^{-1}$ | ignored | N |

```
BNGL

begin parameters
  # provide diffusion constant for used molecule species
  MCELL_DIFFUSION_CONSTANT_3D_A 1.0e-6
  MCELL_DIFFUSION_CONSTANT_3D_B 2.0e-6
  MCELL_DIFFUSION_CONSTANT_3D_C 1.3e-6
end parameters

begin compartments
  # 3D (volume) compartment with volume 1um^3
  CP 3 1
end compartments

begin seed species
  # release 100 molecules of A and 100 of B in compartment CP
  A@CP 100
  B@CP 100
end seed species

begin reaction rules
  # a simple rule for reaction between A and B creating C as the
      product
  # the reaction rate constant is assumed to be in units um^3*1/N*1/s
  A + B -> C 100
end reaction rules
```

**Fig 10. Example model using BNGL.** BNGL file that defines a compartment CP, and instantiates release of 100 molecules of A and 100 molecules of B into it. It then implements a reaction rule in which A and B react to form the product C.

Note that the file in Fig 10 is a standard BNGL file that can be used directly by other tools such as BioNetGen so that no extra conversion steps are needed for the BNGL file to be used elsewhere. This permits fast validation of a reaction network with BioNetGen's ODE or other solvers. The model can be checked against the spatial simulation results in MCell4 without the need to have multiple representations of the same model.

## 3 Results

### 3.1 Testing & validation

We performed extensive testing and validation to ensure the accuracy of results generated by MCell4. The MCell source code repositories include a validation test suite containing more than 350 tests, 45 of which have exact analytic solutions, developed and refined over decades. All versions of MCell are tested using this test suite which stands as a single point of reference for correct results. Here we compare results from the previous versions, MCell3 [4] and MCell3-R [15], which were themselves extensively tested prior to their release. One can obtain

```
MCell4 Python API

import mcell as m

model = m.Model()

# specify that this model uses BioNetGen units (see Table 1)
model.config.use_bng_units = True

# load the information on species (diffusion constants),
# reaction rules, also creates compartment CP as a box with
# volume 1um^3 and creates release sites for molecules A and B
model.load_bngl('subsystem.bngl')

model.initialize()                    # initialize simulation state
model.run_iterations(10)              # simulate 10 iterations
model.end_simulation()                # final simulation step
```

**Fig 11. Example of loading and running BNGL model in MCell4.** Python code for an MCell4 model that will implement loading of the BNGL file shown in Fig 10 (referenced as subsystem.bngl). In this example the entire BNGL file is read. It is also possible to load only specific parts of the BNGL file, for example only reaction rules or only compartment and molecule release information. It is also possible to replace BNGL compartments with actual 3D geometry.

byte for byte identical results with MCell3/MCell3-R and MCell4 by defining specific C/C+ + macros during compilation of the MCell code. These compilation options ensure that the molecules are simulated in the same order and with the same stream of random numbers in MCell3/MCell3-R and MCell4. When applied to the test suite, we obtain byte for byte identical results comparing MCell3, MCell3-R, and MCell4. Simulation results were also validated against results with a BioNetGen ODE solver [6] and with NFsim [12] by running equivalent models in MCell4 and in BioNetGen, running with up to 1024 different random seeds. For valid quantitative comparison the diffusion constants in MCell4 were set to a high value to emulate a well-mixed solution. We then compared the shape of the time course of the averaged counts (and variance of counts) of molecules of a given species. We did not encounter any statistically significant differences between MCell3, MCell3-R, MCell4 and BioNetGen under the well-mixed assumption mentioned above. Also, the results of all 45 tests having analytic solutions agreed well between simulators, and were always within the measured variance in all cases. Some of these tests are referenced as examples in MCell4's API reference manual [26]. All models used in this Testing and Validation section, including simulation output files and scripts used to analyze and plot the data, are available in the GitHub repository for this article [37].

**3.1.1 SNARE complex.** We implemented a model of the SNARE complex, a cooperative dual $Ca^{2+}$ sensor model for neurotransmitter release [38], as an example of an MCell4 model containing a BioNetGen specification. The model includes the binding of up to five calcium ions to the sensor and synchronous or asynchronous modes of release of neurotransmitters. An adapted version of this model was previously implemented in an older version of MCell [39]. The model is composed of SNARE complexes having 36 possible states, calcium ions,

and 126 reactions. There are different possible implementations of the model in BNGL. The one presented here is compatible with MCell4, and allows simulation of the model in BioNet-Gen and MCell4 without modifying the code. It consists of three molecule types and ten reaction rules (Fig 12). The snare complex (represented as **snare**) is an elementary molecule that has 3 components: component **s** with 6 states, that represent the binding site for 0–5 calcium ions in the synchronous sensor; component **a** with 3 states that represent the binding sites for 0–2 calcium ions in the asynchronous sensor; and one component called **dv** with two states (denoted $\sim 0$ and $\sim 1$ in BNG notation), that represents docking of a vesicle to the snare complex ($\sim 1$) or its absence ($\sim 0$). Calcium ions ($Ca^{2+}$) can bind and unbind to the **s** or **a**

```
BNGL

begin compartments
    # Plasma membrane (PM) 2D compartment with volume 0.01 um x SA um^2
    PM 2 6e-4
    # Cytoplasm (CP) 3D volume compartment with volume 1e-3um^3
    CP 3 1e-3 PM
end compartments

begin molecule types
    snare(s~0~1~2~3~4~5,a~0~1~2,dv~0~1)
    Ca
    V_release()
end molecule types

begin species
    # SNARE complex are released in the PM
    snare(s~0,a~0,dv~1)@PM 70
    # Fixed calcium number in the cytosol
    Ca@CP Ca0
end species

begin observables
    Molecules SNARE_sync snare(s~5)
    Molecules SNARE_async snare(a~2)
    Molecules V_release V_release()
end observables

begin reaction rules
    # Calcium binding to the synchronous component of the sensor
    snare(s~0)@PM + Ca@CP <-> snare(s~1)@PM 5*ksp, 1*b^0*ksm
    snare(s~1)@PM + Ca@CP <-> snare(s~2)@PM 4*ksp, 2*b^1*ksm
    snare(s~2)@PM + Ca@CP <-> snare(s~3)@PM 3*ksp, 3*b^2*ksm
    snare(s~3)@PM + Ca@CP <-> snare(s~4)@PM 2*ksp, 4*b^3*ksm
    snare(s~4)@PM + Ca@CP <-> snare(s~5)@PM 1*ksp, 5*b^4*ksm

    # Calcium binding to asynchronous component of the sensor
    snare(a~0)@PM + Ca@CP <-> snare(a~1)@PM 2*kap, 1*b^0*kam
    snare(a~1)@PM + Ca@CP <-> snare(a~2)@PM 1*kap, 2*b^1*kam

    # Synchronous vesicle release
    sync: snare(s~5,dv~1)@PM -> snare(s~5,dv~0)@PM + V_release()@CP gamma
    # Asynchronous vesicle release
    async: snare(dv~1,a~2)@PM -> snare(dv~0,a~2)@PM + V_release()@CP a*
        gamma
    # Vesicle docking to SNARE
    snare(dv~0) -> snare(dv~1) k_dock
  end reaction rules

end model
```

**Fig 12. Compartmental BNGL implementation of the SNARE complex model.** One 3D compartment, cytosol (CP), and its associated plasma membrane (PM) are defined. Molecule types are defined, and their released sites are specified: SNARE molecules are released into the PM, and Calcium ions into the Cytosol. This code is followed by specification of the observables, and the reaction rules governing the interactions.

components of the complex. The release of neurotransmitters is tracked via a dummy molecule type called V_release(), which captures the timing of the release but does not actually release molecules of neurotransmitter (see the next section for an implementation of the release in MCell4). Fig 13A shows code implementing the states of the model, and the synchronous and asynchronous release. Assuming well-mixed conditions, a large volume containing the surface complexes, a large number of complexes, and a constant calcium concentration, the results obtained with BioNetGen ODE simulations and the spatial model in MCell4 give qualitatively similar results (Fig 13B). The source code for this example can be found in [37].

**3.1.2 Event-driven release of neurotransmitter by the SNARE complex.** To release neurotransmitters in an event-driven manner at the times captured by observing V_release() in the SNARE example above, we employ a new feature in MCell4: callbacks. One of the most powerful new features of MCell4 is the ability to implement Python code to be executed (i.e. called) each time a user-specified reaction or surface collision event occurs during a simulation; thus, the term "callback". In the MCell4 Python API the code to be executed when "called" by the event, is written as a function and this function is referred to as a "callback function".

In this case, we created a callback function that will release a given number of neurotransmitter molecules, at the time the synchronous or asynchronous reactions occur. We localize the release at the position of the individual SNARE complex that triggers the release. Here we briefly describe how this is accomplished. The full details and Python source code of the working MCell4 model can be found at https://github.com/mcellteam/article_mcell4_1/tree/master/snare_complex/snare_w_callback.

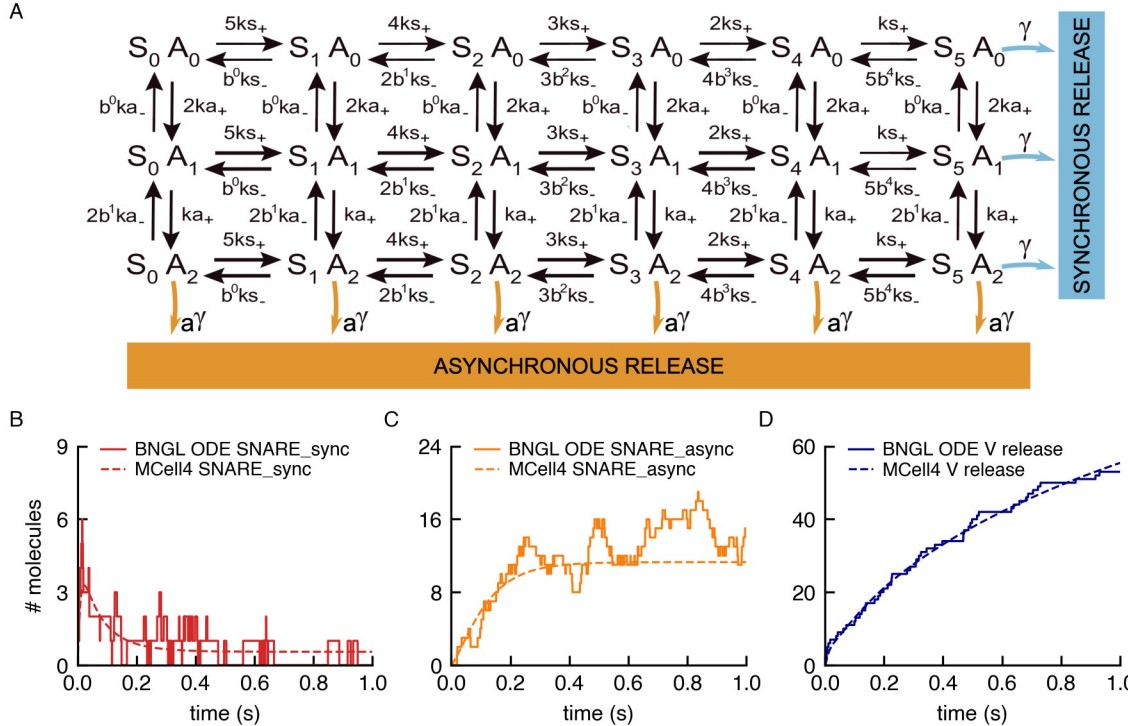

**Fig 13. Validation of model of the SNARE complex.** (A) Schematic diagram of the state variables of the SNARE complex model. It consists of a total of 36 states: 18 with a docked vesicle and 18 without a docked vesicle. The 2 separate vesicle states are not shown in the diagram. S and A represent the synchronous and asynchronous components of the complex, which can be in 6 and 3 different states respectively. (B-D) Results of independent simulations of the model with ODEs in BioNetGen (dashed lines) and in MCell4 (solid lines).

There are two types of callback functions supported in the MCell4 Python API, "reaction callback functions" and "wall hit callback functions" (surface hit). In the SNARE complex example we have created a reaction callback function that will be called upon the stochastic occurrence of the "sync" or "async" reactions specified in Fig 12. We name this reaction callback function "release_event_callback" and associate it with the reactions using the "register_reaction_callback()" command provided in the API. During simulation of the model, whenever a sync or async reaction occurs, the MCell4 physics kernel will execute "release_event_callback()". To specify which species of neurotransmitter to release, how much, and where the register_reaction_callback() command allows additional metadata (called "context") to be passed to the callback function. In this example, we created a Python class called "ReleaseEventCallbackContext" which contains the name of the species and number to be released, as well as the relative release location. Release_event_callback() can then make use of this context to perform the desired operations. See file "customization.py" in the working model for complete details. An MCell4 model with callbacks cannot be exported to an equivalent BNGL representation.

**3.1.3 CaMKII model with large reaction network.** To demonstrate results for a system with a large reaction networks, we use a model of a CaMKII dodecamer which is an extension of a model described in [40].

The CaMKII dodecamer (a "protein complex") is composed of two CaMKII hexameric rings stacked on top of each other. Each CaMKII monomer with its calmodulin (CaM) binding site can be in one of 18 states. The total number of states possible for a CaMKII dodecamer CaMKII is then $18^{12}/12 \approx 10^{14}$ (the division by 12 is to remove symmetric states). This is an example of the combinatorial complexity mentioned in section 2.4 for which it is simply not feasible to expand all the reaction rules and generate the entire reaction network to be stored in memory, and thus a network-free approach is necessary. Fig 14 shows the results of validation of this model against BioNetGen/NFsim, MCell3R, and MCell4.

We also present an extension of the aforementioned model [40], in which we can now observe the effects of the geometry of the compartment on the simulation results by modeling in MCell4. Fig 15 shows three different variations of the model. The first variation distributes the molecules homogeneously in the compartment (equivalent to the well-mixed versioned

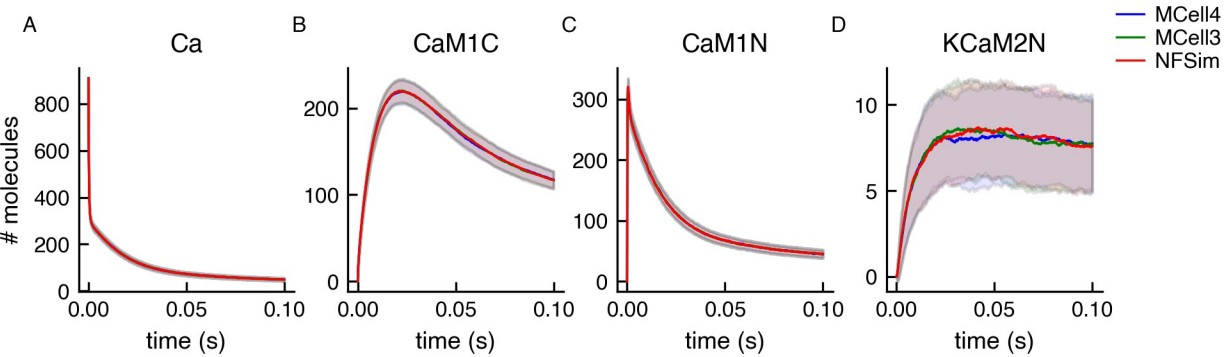

**Fig 14. Validation of MCell4 simulation against BioNetGen/NFsim and MCell3R using a model of CaMKII.** The input BNGL model for NFsim was obtained by automatic BNGL export of BNGL from the MCell4 model. The simulation ran for 100000 iterations (0.1 s). Lines in the graphs are averages from 256 runs with different random seeds, and bands represent one standard deviation. Molecules in MCell3R and MCell4 use diffusion constant of $10^{-3} cm^2/s$ to emulate a well-mixed solution (the usual value is around $10^{-6} cm^2/s$). The names of the observed species are indicated in the graph titles: A) Ca is unbound calcium ions; B) CaM1C is CaM(C $\sim$ 1, N $\sim$ 0, camkii); C) CaM1N is CaM(C $\sim$ 0, N $\sim$ 1, camkii); D) KCaM2N is CaMKII(T286 $\sim$ U, cam!1).CaM(C $\sim$ 0, N $\sim$ 2, camkii!1). The simulation was initiated far from equilibrium; therefore there was an initial jump in the molecule numbers. The molecule names are explained in [40].

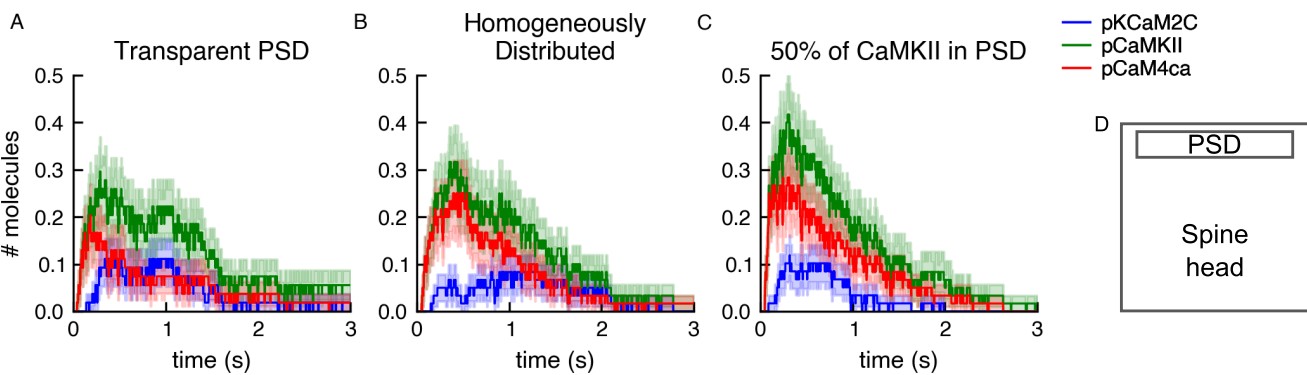

**Fig 15. The effect on CaMKII phosphorylation of trapping CaMKII and CaM inside the PSD of a dendritic spine head.** Three different conditions were simulated within a compartment representing a dendritic spine head containing a subcompartment representing a postsynaptic density (PSD) shown in panel D. (A) All molecules are homogeneously distributed throughout the entire spine head. The PSD is made to be transparent to the diffusion of all molecules. (B) The PSD is made to be reflective to CaMKII and CaM, but transparent to calcium ions and PP1. All the molecules are homogeneously distributed throughout both compartments. (C) The PSD is made to be reflective to CaMKII and CaM and 50% of the CaMKII molecules are trapped inside the PSD, while the rest of the molecules are distributed homogeneously throughout the remainder of the spine head. D) Diagram of the spine head compartment 0.5 x 0.5 x 0.5 um in size containing the PSD subcompartment 0.05 x 0.4 x 0.4 um in size. The plots show an average of 60 runs, lighter-shaded bands represent the standard error of the mean.

published previously, Fig 15A). Two additional variations include a small subcompartment, located near the top of the larger compartment, that is not transparent to diffusion of CaMKII and CaM molecules. In the first variation all the molecules are homogeneously distributed throughout the compartment, but the CaMKII and CaM molecules in the subcompartment do not mix with the rest of the compartment (Fig 15B). In the second variation, half of the CaMKII molecules are placed in the subcompartment and the other half in the remainder of the compartment, while CaM is still distributed homogeneously throughout the entire volume (Fig 15C).

We sought to observe the effect of these three conditions on CaMKII phosphorylation as a result of $Ca^{2+}$ influx into the compartment. In all three conditions a $Ca^{2+}$ influx is simulated from a single point source located in the center of the top face of the large compartment. As in [40] the $Ca^{2+}$ influx was such that at the peak the free calcium concentration was $\sim 10\mu M$, and it returned to near the steady state level within 100 ms. These spatial differences have a small but significant effect on CaMKII phosphorylation levels in response to the $Ca^{2+}$ influx. These differences would have been impossible to investigate without the combination of the network-free simulations and the diffusion in space implemented in MCell4.

**3.1.4 Volume-surface and surface-surface reactions: Membrane localization model.** We used a membrane localization model (from [41], section 2A) to validate volume-surface and surface-surface reactions. The model illustrates how membrane localization stabilizes protein-protein interactions [42]. A pair of protein binding partners A and B, diffusing in the volume, are localized to the membrane surface by binding a lipid molecule M in the membrane, to form molecules of MA and MB. This binding to the membrane constrains the space in which molecules of MA and MB diffuse and thus promotes further complex formation between MA and MB within the membrane. The model is created within a box of dimensions $0.47 \times 0.47 \times 5\mu m^3$. Surface molecules, M, are released on one of the smaller sides of the box. The 4 edges of this side of the box are set to be reflective to 2D diffusion so the surface molecules cannot diffuse onto the other sides. Volume molecules A and B are released inside the box.

MCell subdivides the surface areas of geometric objects into small tiles. A maximum of one molecule can occupy one tile at a time—this tiling simulates volume exclusion for surface

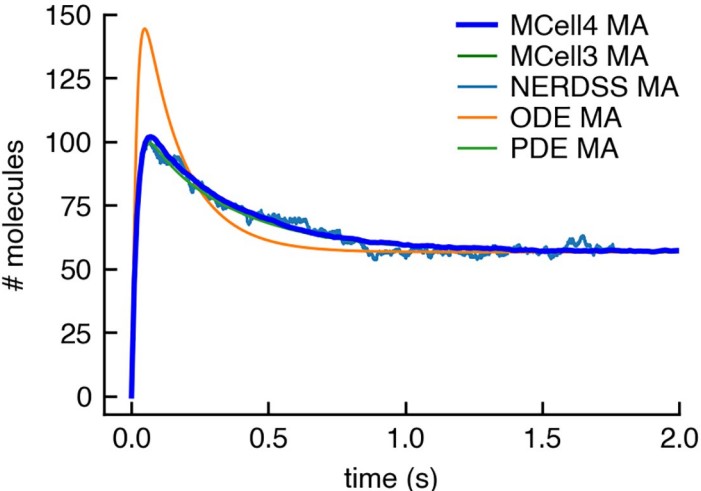

**Fig 16. Simulation of membrane localization.** The plot shows copy numbers of a surface molecule MA (surface molecule M with a bound volume molecule A). MCell4 and MCell3 results show a good match with simulations using the NERDSS simulator (NERDSS results are from from [41]) and with simulations in VCell using a PDE method (VCell results are from from [41]). Results of simulations with an ODE method in VCell differ from MCell3, MCell4, NERDSS, and PDE solutions. MCell3 and MCell4 results are an average of 512 runs with different random seeds.

molecules. A simulation parameter named "surface grid density" sets the density (and size) of the tiles and thus the maximum packing density of surface molecules. The initial density of surface molecules, M, in this model is 17000 $molecules/\mu m^2$ (i.e. 3755 molecules total), and we set the surface grid density to 40000 $tiles/\mu m^2$. This results in an initial occupied area fraction of 17000/40000, or 42.5%, leaving room for diffusion of the surface molecules. The results of simulations comparing MCell4 to MCell3, NERDSS (NonEquilibrium Reaction-Diffusion Self-assembly Simulator [43], a spatial stochastic simulator), VCell (virtual cell [44, 45]) using PDEs, and VCell using ODEs are shown in Fig 16. MCell4 demonstrated excellent agreement with all simulation methods with the exception of the well-mixed ODE method, which is to be expected for such a spatially inhomogeneous system involving 3D and 2D reaction-diffusion. This model demonstrates that membrane localization can affect the speed and stability of protein complex formation and that this behavior cannot be modeled precisely with well-mixed ODE methods.

**3.1.5 Stochastic fluctuations in a system with multiple steady states: Autophosphorylation.** Another validation model from [41] (section 2B) shows that positive feedback induces stochastic switching in a kinase autophosphorylation circuit which is a a system with multiple steady states. A deterministic ODE solution does not show these multiple steady states and almost immediately stabilizes in one of them. In Fig 17 we show the output of an MCell4 simulation and a simulation of the same model simulated in NFsim using the BNGL exported from the MCell4 model (more details on BNGL export are in 2.4.2). We also illustrate the steady states reached with ODE solutions.

## 3.2 Performance

With relatively small reaction networks (less than 100 or so reactions), the performance of MCell4 is similar to MCell3 as shown in Fig 18(A). MCell3 is already highly optimized. MCell3 contains optimization of cache performance that speeds up models with large geometries; this optimization is not present in MCell4. Thus MCell3 is faster for large geometric

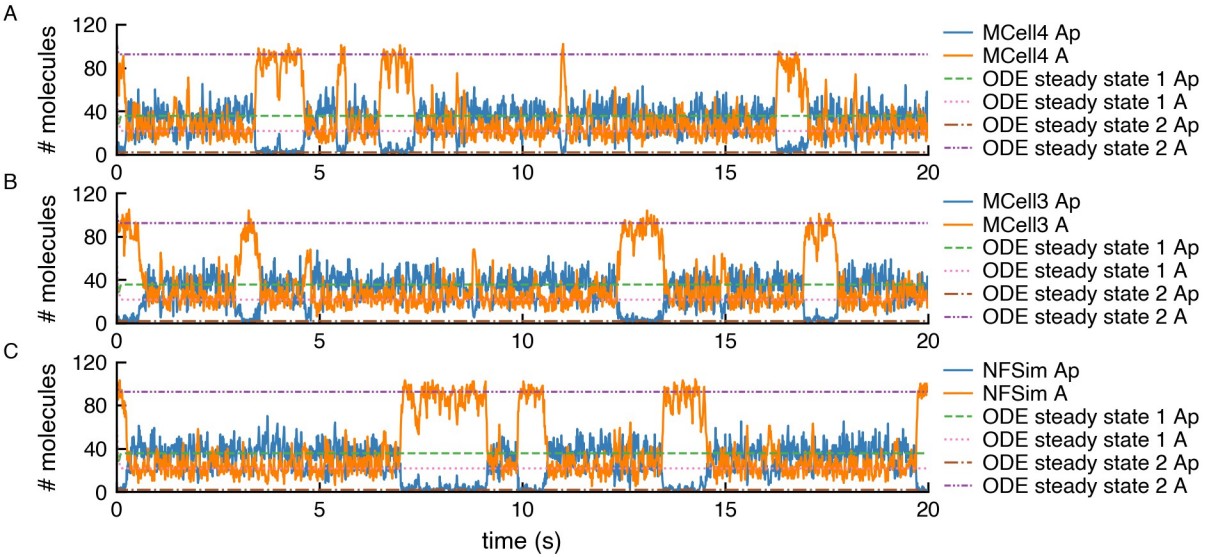

**Fig 17. Simulation of a system that exhibits stochastic switching between multiple steady states.** Copy numbers of unphosphorylated kinase A and its phosphorylated variant Ap are shown for a single simulation run in: A) MCell4; B) MCell3; and C) NFsim. The NFsim model was obtained by automatically exporting the MCell4 model into BNGL. The graphs also show solutions obtained with a deterministic ODE model for which data from [41] were used. The results demonstrate that the MCell results correctly reach one of the stable steady states shown in the ODE results. The simulation stays in such a state, and then due to stochastic behavior, a switch another steady state occurs.

models such as models created in neuropil reconstructions containing on the order of 4 million triangles defining their geometry [5]. The situation is different when comparing MCell4 and MCell3-R with models that use large BNGL-defined reaction networks (Fig 18B). MCell3-R uses the NFsim library to compute reaction products for BNG reactions. With large

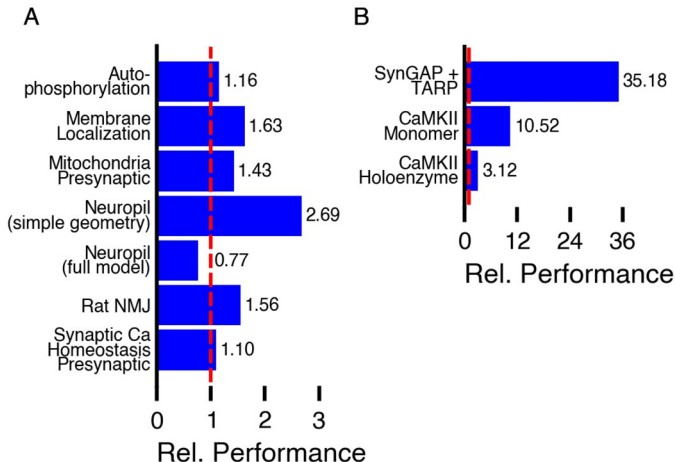

**Fig 18. Performance benchmarks of MCell4.** For selected benchmarks, we measured elapsed time for how long the simulation ran starting from the second iteration (after all initializations) and ending when the simulation finished. Time was measured on AMD Ryzen 9 3900X@3.8GHz. Both MCell3 and MCell4 use a single execution thread. Relative performance shown in the graphs is computed as time for MCell3 (panel A) or MCell3-R (panel B) divided by time for MCell4. The sources of the models are as follows: Presynaptic Ca Homeostasis [39]; Rat Neuromuscular Junction [2] model with updated geometry (shown in Fig 1), Neuropil [5]; Mitochondrion Model [47]; Membrane Localization [41]; Autophosphorylation [41]; CaMKII Monomers [40]; CaMKII Holoenzyme [40]; SynGAP and TARP with the PDZ domains of PSD-95 (MCell4 model not yet published).

reaction networks containing as many as $10^{10}$ reactions or more, MCell3-R stores all the reactions that have occurred during run-time in memory and thus gradually slows down. We have not been able to implement reaction cache cleanup in MCell3-R. MCell4 with the BNG library keeps track of the number of molecules of each species in the system during simulation and periodically removes from the cache reactions and species that are no longer in use. This facilitates simulation of complex reaction networks with a potentially infinite number of species and reactions without excessive impact on memory usage and performance. In comparison to MCell3-R, with MCell4 we observed a speed-up of 10x for the CaMKII holoenzyme model (having potentially $10^{14}$ states) and a speed-up of 35x for a model of multimeric interations of SynGAP and TARP with the PDZ domains of PSD-95 forming irregular condensates potentially limitless in size and number of states [46] (MCell4 model not yet published). Execution times were measured on a Linux workstation with AMD Ryzen 9 3900X processors @3.8GHz, Debian 10 operating system. MCell4 was compiled with gcc 9.3, at optimization level -O3, for use with Python 3.9.

## 3.3 Hybrid simulation of a circadian clock

MCell4's Python API supports interaction with an MCell4 simulation while it is running. Here we show a model in which the progression over time of one molecular species is implemented in Python code as an ordinary differential equation and the remaining species are implemented in MCell4 as particles behaving stochastically, with coupling and interactions between the two implementations. As a basis for this demonstration we used a model of a circadian clock published in article [41], originally based on article [48]. This section serves mainly as a demonstration of selected MCell4 features. A more circumspect treatment of the general domain of hybrid simulations is out of the scope of this article and we kindly refer interested readers to references [49], [50], [51], [52] or [53]. We note however, that the Python API in MCell4 was designed to be fine grained and is compatible with external physics engines, hybrid methods and strategies developed elsewhere.

The model simulates the behavior of an activator protein A and repressor protein R that are produced from mRNA transcribed from a single copy of a gene (one for each protein). Coupling of A and R expression is driven by positive feedback of the activator A, which binds to each gene's promoters to enhance transcription. Protein R also binds to A to degrade it. All other proteins and mRNA are degraded spontaneously at a constant rate.

Compared to the original model in [48], authors of [41] increased the reaction rates in the model from hours to seconds by multiplying the reaction rates by 3600. Because the purpose of this example is to demonstrate a hybrid model in MCell4 and its validation, which requires many runs, we made another change to accelerate the simulation; we reduced the simulation volume by a factor of 268 to 0.25 μm which increased the rate of bimolecular reactions. We also increased the unimolecular reaction rates by the same factor.

In the hybrid model, we chose protein R to be simulated as a changing concentration, under well-mixed conditions, whose concentration value is updated by finite difference expressions. The other species are simulated as particles. We could have chosen any of the species to treat as concentrations rather than particles. But in doing so one is making making the important assumption that those species remain well-mixed at all times through out the simulation volume. If the well-mixed assumption is valid for the chosen species a hybrid model can be used as a powerful means to speed up the execution time of a model, especially if the well-mixed species exist in large numbers.

In the base MCell4 model, there are 4 reactions that consume or produce R (Fig 19). We replaced two of these with reactions that do not model R as a particle. The remaining two

**BNGL Reactions**

```
A_and_R_to_AR:            A + R -> AR          AR_kon      # 1/M*1/s
R_to_0:                   R -> 0               R_koff      # 1/s
mRNA_R_to_mRNA_R_plus_R:  mRNA_R -> mRNA_R + R mRNA_R_koff # 1/s
AR_to_R:                  AR -> R              AR_koff     # 1/s
```

**Fig 19. Reaction rules affecting repressor protein R in the particle-only model.**

reactions were replaced with finite difference expressions (Fig 20). The hybrid coupling of the finite difference calculations with MCell4's particle-based calculations is shown in the pseudo-code representing the main simulation loop in Fig 21. Note that the bimolecular reaction labeled 'A_and_R_to_AR' in the particle only model has been transformed into the pseudo first-order reaction labeled 'A_to_R' in the hybrid model where the pseudo first-order rate A_koff is updated during step 3 of the the simulation loop.

To validate that the results of the hybrid variant of the model are correct, we ran 1024 instances of stochastic simulations with different initial random seeds. We also compared the effect of two different diffusion constant values when using MCell4. Results reporting the average oscillation frequencies are shown in Fig 22 and the copy numbers of molecules A and R in Fig 23.

When using a fast diffusion constant of $10^{-7}$ $cm^2/s$ for all molecules, all simulation approaches produce essentially the same results. A significant advantage of using hybrid modeling is often that the hybrid model runs much faster, as in this specific example, in which the simulation speed of the MCell4 hybrid model is 4x faster. This is because: 1) The time step can be set to 5x longer because there is no need to model explicitly the diffusion of R as particle-based molecules for the fastest reactions involving species R. Note that the time step when all molecules are modeled as particles must be $10^{-7}$ $s$ to accurately model these fast reactions and keep the probability of reacton per collision to be less that 1 (as noted in section 1.1); 2) species R is not modeled as particles and thus there is no need to compute the diffusion trajectories of each molecule of R.

Interestingly, we find that the oscillation frequency increases, and the standard deviation of the frequency decreases, when the diffusion constants of the particles are decreased. This effect is greatest in the non-hybrid MCell4 pure particle model. This is perhaps non-intuitive but can be explained by localized concentration gradients that form, when diffusion is slow, near the

**BNGL Reactions**

```
A_to_AR:                  A -> AR              A_koff      # 1/s
# R_to_0:                 - modeled as ODE
# mRNA_R_to_mRNA_R_plus_R: - modeled as ODE
AR_to_0:                  AR -> 0              AR_koff     # 1/s
```

**Fig 20. Reaction rules affecting repressor protein R in the hybrid model.**

```
MCell4 Pseudo-code

num_R = 0.0                     # in N, initial copy number of Rs,
                                # modeled as a floating-point value

T_STEP = 5e-7                   # in us, simulation time step
NA = 6.0221409e+23              # in N/mol, Avogardo's constant
VOLUME = 4.188993 * 1e-15       # in l, simulated volume

for i in range(ITERATIONS):
    # 1) Run particle-based simulation for 1 time step
    model.run_iterations(1)

    # 2)  Update the concentration-based copy number of Rs
    # 2.1) Rs consumed by original reaction A + R -> AR
    dR_due_A_to_AR =
        -model.get_number_of_reactions_in_last_iteration('A_to_AR')

    # 2.2) Rs consumed by original reaction R -> 0
    dR_due_R_to_0  =
        -(num_R * R_koff * TIME_STEP)

    # 2.3) Rs produced by original reaction mRNA_R -> mRNA_R + R
    dR_due_mRNA_R  =
        model.get_number_of_molecules('mRNA_R') * mRNA_R_koff * T_STEP

    # 2.4) Rs produced by original reaction AR -> R
    dR_due_AR_to_0 =
        model.get_number_of_reactions_in_last_iteration('AR_to_0')

    # 2.5) Update the copy number of Rs
    num_R +=
        dR_due_A_to_AR + dR_due_R_to_0 + dR_due_mRNA_R + dR_due_AR_to_0

    # 3) Update rate of reaction A -> AR (originally A + R -> AR):
    # Sets the rate A_koff using concentration of R effectively
    # converting a bimolecular reaction rate from units of 1/M*1/s to a
    # pseudo first-order unimolecular rate in units of 1/s.
    # Concentration is here computed with copy number of Rs
    # truncated to the closest integer to avoid reactions happening
    # when there is less than 1.0 Rs.
    concentration_R = floor(numR) / NA / VOLUME   # in 1/M
    model.set_reaction_rate('A_to_AR', concentration_R * AR_kon)
```

**Fig 21. Pseudo-code of the main simulation loop.** This script in pseudo-code: 1) runs an iteration of the particle-based simulation, 2) updates the copy number of R based on the current MCell4 state, and 3) updates the rate of reaction A -> AR that was originally a bimolecular reaction A + R -> AR. N is a unit representing the copy number. This pseudo-code was adapted to show the actual computations in a more comprehensible way. The runnable MCell4 Python code is available in the GitHub repository accompanying this article [37].

single copies of the genes for A and R, concentrating particles of A (and also R, in the pure particle model) together with the mRNAs for A and R along with their reaction products, as they bind and unbind. Thus small nanodomains form increasing the rate of reaction, and decreasing the standard deviation, in these local domains. In a purely well-mixed model (NFsim, SSA) such nanodomains can never exist. In a spatial particle model with rapid diffusion any nanodomains are rapidly dispersed. Thus we can predict that for slowly diffusing molecules of A, R and mRNA (and the two genes) the well-mixed assumption is likely violated and a non-hybrid spatial particle method is recommended.

This is a relatively simple example in which we compute the ODE separately with Python code, however it shows the strength of this approach in which one can couple other physics engines to MCell4 and achieve multi-scale simulations.

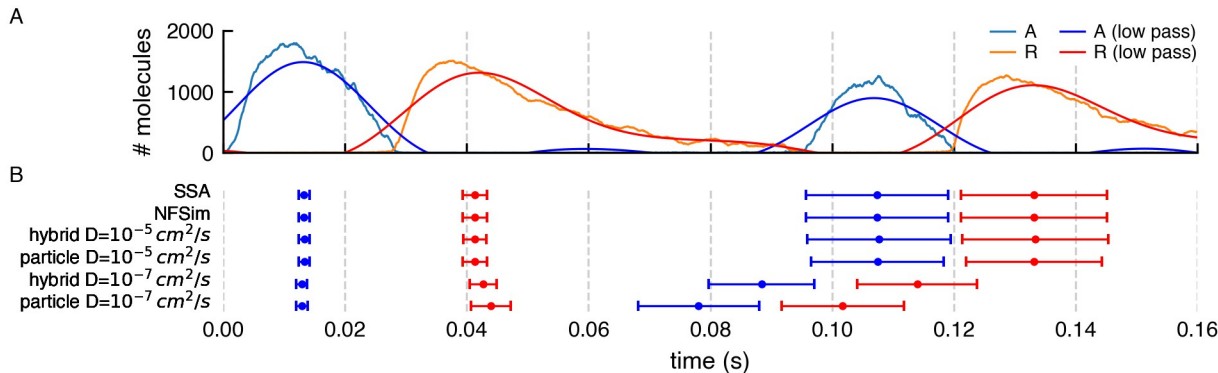

**Fig 22. Simulation of a circiadian clock.** (A) Result of a stochastic simulation of a circadian clock model with NFsim. Copy numbers of molecules A and R show periodic oscillation. A low pass frequency filter was used to smooth the values of A and R. The reason for the smoothing was to get a numerical value related to the actual peak. The peaks from low-pass filtered data do not represent true average peaks but can be used as a proxy to obtain the time of a peak for comparison with other simulation methods. (B) The error bars capture the mean and standard deviation of the low pass filtered peak times for different variants of the model and simulation algorithms. Each of the variants was run 1024 times. It is evident that the SSA, the NFsim, and the MCell4 model variants with a fast diffusion constant, $D = 10^{-5}\ cm^2/s$, give essentially the same results. The hybrid MCell4 model with the slower diffusion constant, $D = 10^{-7}\ cm^2/s$, shows faster oscillation than the non-spatial models run with SSA and NFsim, and the MCell4 variants with faster diffusion. The pure particle-based MCell4 model with $D = 10^{-7} cm^2/s$ shows the fastest oscillations.

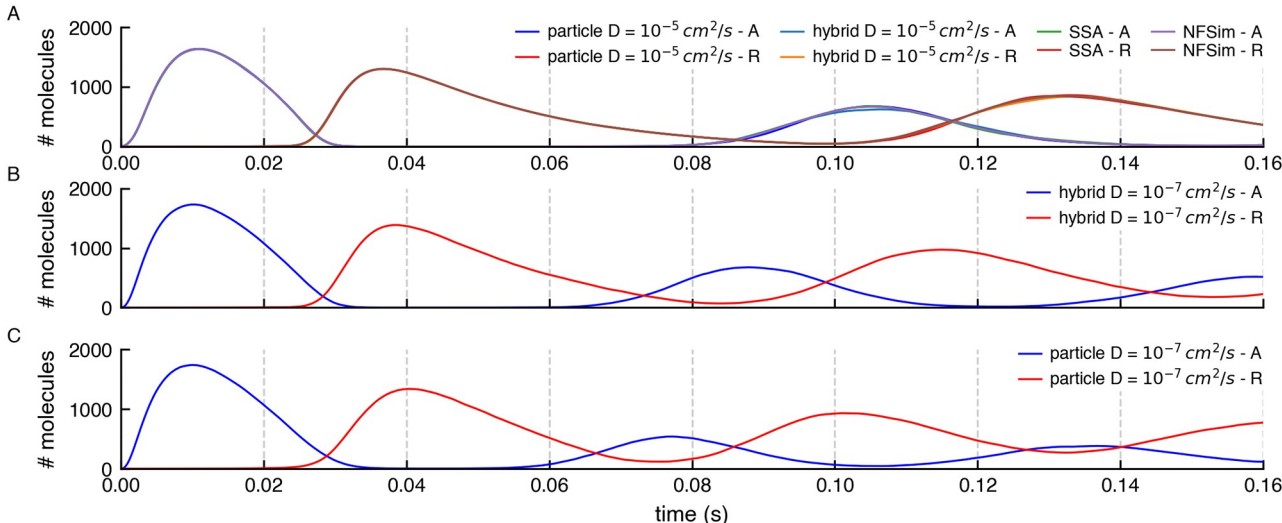

**Fig 23. Comparison of copy numbers of A and R during simulations by different methods.** (A) The average copy numbers for A and R proteins from 1024 runs in NFsim, SSA, and MCell4 with a fast diffusion constant match. To get an even better match, would require more than 1024 runs because stochastic molecular simulations show high variability when the copy number of some of the species is low which is the case here for both A and R. (B) and (C) Average copy numbers for MCell4 simulations with a slow diffusion constant. These are shown as separate plots to highlight the effect of slow diffusion on spatial simulation results.

# 4 Conclusions

## 4.1 Summary

We have described MCell4, a newly updated particle-based reaction-diffusion tool based on Monte Carlo algorithms that allows spatially realistic simulation of volume and surface molecules in a detailed 3D geometry. MCell4 builds on features of MCell3 (and MCell3-R),

providing improved integration with the BioNetGen Language as well as a Python API that enables fine control of a simulation through Python code.

In MCell4, as opposed to MCell3, molecules and reactions are natively written in BNGL allowing a seamless transition between MCell4 and BNG simulation environments. The update has dramatically improved the ability to run network free simulations in the spatial MCell environment, when compared to the previous MCell3-R which employed the NFsim engine to run reaction written in BNGL [12, 15].

The new Python API, enables one to write Python code that can change geometry, reaction rates, create or remove molecules, execute reactions, etc., during a simulation. This powerful new feature allows construction and execution of multi-scale hybrid models and is compatible with previously published hybrid methods [49], [50], [51], [52] or [53].

As we have demonstrated here through examples, MCell4 adds many new features including the ability to create and efficiently simulate fully spatial network-free molecular reaction models within realistic geometry. It adds the ability to switch back and forth easily between MCell4 and BNG environments.

Thorough validation was performed and has shown concordance with MCell3, MCell3-R, and BioNetGen in ODE and NFsim modes. Section 3.1.4 also shows a model where MCell4 gives the same results as spatial simulations using NERDSS [43] and VCell PDE method [45]. And the hybrid model of a circadian clock (Section 3.3) in MCell4 demonstrated close agreement with stochastic well-mixed methods (SSA and NFSim), but only when the well-mixed assumption is valid. Only through the use of spatial stochastic particle methods, such as MCell4, can the validity of the well-mixed assumption be rigorously tested in biological systems like this one.

MCell4 is a significant improvement on the previous MCell3-R version with respect to simulation speed (which is about 10x faster for models with large reaction networks), number of features, as well as usability. Together, these improvements allow simulation of new classes of systems that could not be modeled previously, especially systems that exhibit combinatorial complexity and stochastic fluctuations within nanodomains of spatially-resolved particles.

## 4.2 Availability, limitations, and future directions

MCell4 is available under the MIT license. For easy installation and usage, a package containing MCell, Blender, the Blender plugin CellBlender, and other tools are available along with detailed documentation and online tutorials at [54]. MCell4 includes a new C++ library for parsing the BioNetGen language and provides methods to process BioNetGen reactions. This library libBNG is also available under the MIT license [19].

MCell4 does not currently support definition of the spatial shapes of assembled molecular BNGL complexes that could be useful, for instance, when modeling the post-synaptic density [5] or actin filament networks [55] where simply replacing these polymers with a single point in space is inadequate. Furthermore, the ability to model volume exclusion by individual molecules and complexes will be an important goal for the future. We have plans to provide built-in support for combining particle-based simulation with concentration or well-mixed simulation algorithms such as SSA [35] or the finite element method that uses PDEs (partial differential equations), e.g., [45]. Such hybrid modeling will provide means to simulate longer timescales while still being spatially accurate and able to correctly handle cases when the copy number of molecules is low. All these features will be the focus of future developments.

## Supporting information

**S1 Text. Supporting information text.**
(PDF)

## Acknowledgments

The authors thank Dr. Padmini Rangamani for discussions on boundary conditions and the biophysics of diffusion near membranes. We heartily thank Dr. Markus Dittrich, Dr. Burak Kaynak, Dr. Oliver Ernst, Dr. Rex Kerr, Jacob Czech, Jed Wing, Don Spencer, and Robert Kuczewski whose insights on API design for discrete event simulation have guided the development of MCell over the years, laying the foundation for MCell4. We thank Jorge Aldana for his expert technical support of the computing infrastructure in the Computational Neurobiology Laboratory at Salk.

## Author Contributions

**Conceptualization:** Adam Husar, Mariam Ordyan, Guadalupe C. Garcia, Joel G. Yancey, Ali S. Saglam, James R. Faeder, Thomas M. Bartol, Terrence J. Sejnowski.

**Data curation:** Adam Husar, Mariam Ordyan, Guadalupe C. Garcia, Joel G. Yancey, Ali S. Saglam, James R. Faeder, Thomas M. Bartol, Mary B. Kennedy.

**Formal analysis:** Adam Husar, Mariam Ordyan, Guadalupe C. Garcia, Joel G. Yancey, Ali S. Saglam, James R. Faeder, Thomas M. Bartol, Mary B. Kennedy, Terrence J. Sejnowski.

**Funding acquisition:** James R. Faeder, Thomas M. Bartol, Mary B. Kennedy, Terrence J. Sejnowski.

**Investigation:** Adam Husar, Mariam Ordyan, Guadalupe C. Garcia, Ali S. Saglam, James R. Faeder, Thomas M. Bartol, Mary B. Kennedy, Terrence J. Sejnowski.

**Methodology:** Adam Husar, Mariam Ordyan, Guadalupe C. Garcia, Joel G. Yancey, Ali S. Saglam, James R. Faeder, Thomas M. Bartol, Mary B. Kennedy, Terrence J. Sejnowski.

**Project administration:** Terrence J. Sejnowski.

**Resources:** Thomas M. Bartol.

**Software:** Adam Husar, Mariam Ordyan, Guadalupe C. Garcia, Joel G. Yancey, Ali S. Saglam, James R. Faeder, Thomas M. Bartol.

**Supervision:** James R. Faeder, Thomas M. Bartol, Terrence J. Sejnowski.

**Validation:** Adam Husar, Mariam Ordyan, Guadalupe C. Garcia, Joel G. Yancey, Ali S. Saglam, James R. Faeder, Thomas M. Bartol, Mary B. Kennedy.

**Visualization:** Mariam Ordyan, Guadalupe C. Garcia, Joel G. Yancey, Ali S. Saglam, James R. Faeder, Thomas M. Bartol.

**Writing – original draft:** Adam Husar, Mariam Ordyan, Guadalupe C. Garcia, Thomas M. Bartol.

**Writing – review & editing:** Adam Husar, Mariam Ordyan, Guadalupe C. Garcia, Joel G. Yancey, Ali S. Saglam, James R. Faeder, Thomas M. Bartol, Mary B. Kennedy, Terrence J. Sejnowski.

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
