## [Decision Letter · Decision Letter 0]

26 Jun 2023

Dear Dr. Bartol,

Thank you very much for submitting your manuscript "MCELL4 WITH BIONETGEN A MONTE CARLO SIMULATOR OF RULE-BASED REACTION-DIFFUSION SYSTEMS WITH PYTHON INTERFACE" for consideration at PLOS Computational Biology.

As with all papers reviewed by the journal, your manuscript was reviewed by members of the editorial board and by several independent reviewers. In light of the reviews (below this email), we would like to invite the resubmission of a significantly-revised version that takes into account the reviewers' comments.

Please make sure you properly cite existing related work dealing with spatial and stochastic simulation approaches. 

We cannot make any decision about publication until we have seen the revised manuscript and your response to the reviewers' comments. Your revised manuscript is also likely to be sent to reviewers for further evaluation.

Sincerely,

Martin Meier-Schellersheim

Academic Editor

PLOS Computational Biology

Daniel Beard

Section Editor

PLOS Computational Biology

Reviewer's Responses to Questions

**Comments to the Authors:**

Reviewer #1: This paper describes a new version of MCell, version 4, which is a substantial overhaul of the prior versions, MCell3 and MCell3-R. MCell is a widely-used particle-based cell biology simulator. One major new feature is that the reaction entry method has been changed from MCell’s model description language (MDL) to the BNGL language, which supports rule-based modeling and is widely supported. Another major new feature is a Python application programming interface. Both of these are quite valuable, making this a useful update to the software. The new features are demonstrated in this article through examples for SNARE complexes, CamKII modeling, and others. Although not the primary focus of this paper, it also describes a new library interface for the BioNetGen software, which is another important contribution.

Major issues

Overall, this is important work that deserves to be published. However, it should be noted that nearly all of the capabilities described here have been supported by the Smoldyn software (which works at a similar level of detail) for several years, which is not mentioned in this paper. The BNGL support was described in Andrews, 2017; that paper is cited here, although not in this context (Smoldyn’s BNGL support is less integrated into the software than it is here). Smoldyn’s Python API was described in Singh and Andrews, Bioinformatics, 2021. As with MCell4, Smoldyn’s API also supports callback functions and multiscale simulations (e.g. a simulation was described in the 2021 paper that combined the Smoldyn and MOOSE simulators). In addition, Smoldyn has supported transmembrane interactions since 2015 through the reaction_intersurface statement, which is described in the Smoldyn’s User’s Manual but has not been published elsewhere. The fact that these features are available in Smoldyn does not lessen their importance in MCell, but is something that readers should be made aware of.

Several BioNetGen extensions for spatial modeling were presumably developed in this work but are not described here. In particular, how are diffusion coefficients, species names, and display parameters chosen for newly generated species? Are there situations where it’s unclear whether newly generated species should be volume species or surface species and, if so, how is this handled? Also, does this version have backward compatibility with the comma and apostrophe notation that was used in MCell3-R?

Does this BNGL implementation support generate-first, on-the-fly network expansion, or both?

I didn’t understand section 2.2. I think it means that all of the Python API generation and the YAML code was simply a method for developing the Python API in the first place, and is not relevant for actual model development. Is this correct? Also, how does the documentation fit into this development pipeline? Is this documentation for the end user, or only for software developers? It might also be worth mentioning that this uses the pybind11 library, which isn’t mentioned here but appears in figure 4.

It took me a while to decipher Figure 8. In part, row G is listed between rows C and D. But also, is this figure necessary? I think it just describes how BioNetGen works, which wasn’t part of the work described in this paper.

I was pleased to see the extensive validation that was described in section 3.1. The text doesn’t actually say that MCell4 always agreed with the other simulators, although this is implied. Did it always agree, or were there any notable exceptions? Also, is there some way for a reader to see these results in more detail, such as in the download package or in supplementary information? Are there additional concerns that a user should be aware of?

For the performance testing, it would be nice to see a test of the Michaelis-Menten benchmark model that was described in Andrews et al., PLoS Comp. Biol., 2010. This has become a relatively widely used benchmark, enabling comparison between different simulators. Admittedly, results depend strongly on the computer used, but they’re still useful (or, better, is to compare results with other simulators using the same computer). Alternatively, the authors could run the models that they describe in Figure 18 on some other simulator, again with the goal of comparing performance across different simulators.

Minor issues

The paper says that MCell4 is a new C++ implementation of MCell. Was it really rewritten from scratch? If so, this is a Herculean effort.

I’m curious about the support for the traditional MDL language. Is it only retained for backward compatibility, or will it continue to be developed as new features are added? Also, to what extent will CellBlender be supported, since it’s another easy-to-use option for non-experts.

Grammar or spelling mistakes in lines: 122, 159, 184, 215, 241, 308, 353.

Figure 1 is hard to see.

Reviewed by Steve Andrews

Reviewer #2: Review uploaded as attachment.

Reviewer #3: In their article "MCell4 with BioNetGen: A Monte Carlo Simulatior of Rule-Based Reaction-Diffusion-Systems with Python Interface" the authors Husar et al. deliver a detailed description of the new technical features of the MCell4 simulation environment and exemplify its capibilities by simulating several biochemical models, which at once validate the MCell4 simulations with respect to previous simulation frameworks and ODE models and are used for a comparison of the computational performance of the new framework.

Without any doubt, the authors present a highly useful extension that largely enhances the capabilities of an important biochemical simulation framework, which renders their work highly relevant to the computational biology community and beyond. In particular the combination of MCell with BioNetGen is a major step forward and highly augments the versatility of this simulation tool. The article therefore fits very well into the spectrum of PLoS Computational Biology and would constitute an excellent and interesting contribution once adequately adapted.

I found it a bit harder to come to clearly positive conclusion on the form of presentation, i.e. the way the content is composed and presented. In my opinion, in its current form, the focus of the article is still too technical. The technical extensions are presented in great detail, sometimes up to the point that the names of concrete Python routines are mentioned in the main text. On the other hand, while the authors present a decent range of simulated example systems, the description of these models and in particular of the simulation results is a bit short, sometimes almost being reduced to one sentence mentioning good agreement between MCell4 sims and previous approaches. This part should be expanded and rewritten with a stronger focus on the biological aspects, while details such as the part on specific callback functions used should be outsourced into a supporting text.

Also the organization of the figures should be changed. 23 main figures, of which most are code fragments, seem inappropriate for the format of a PLoS Comp Biol article. Many figures could be grouped together into separate panels of one figure (such as, e.g., Figs. 2 and 3 and Figs. 4--6; btw., please enlarge the arrow heads in these figures). The authors should consider, again, outsourcing some of the figures into a supporting document; some of the figures showing code fragments may be good candidates for this. There is also need for improving some of the captions. For example, the caption of Fig. 18 does not clearly refer to its seperate panels A and B.

The discussion / conclusion section is the weakest part of the article. It is too short, does not talk about the performance and accuracy of the new simulator, and also not about the biological systems that were simulated. It mainly focuses on future directions / extensions. Also there is no comparison between MCell4 and other prominent simulation schemes. What are the advantages / disadvantages of using MCell4 / preferring it above other schemes? This part should be thoroughly rewritten and expanded.

In terms of language, the article is generally well written, but still contains many occurances of stylistic and linguistic shortcomings, such as missing articles or other words, or usage of singular where plural should be used, up to the point that I found it hard to believe that the mother tongue speakers in the author list read the article with the intention of fixing such issues. I will not list these minor issues here because I believe that large parts of the article need to be rewritten with a different focus anyhow. But in a new version of the article, all authors should make an effort in ensuring that the article is well written and uses correct wording. Issues such as missing articles or swapped words should not be fixed by reviewers or copy editors.

I therefore recommend a thorough revision of the article with a particular emphasis on reducing / outsourcing some of the technical details, while rewriting the part on example simulations with a stronger focus on the biological aspects and results (even when these agree with previous results); this would make the article also attractive and more accessible to biologists and experimentalists, while in its current form it mainly targets experts of biochemical simulator development, i.e. mainly computer scientists.

At the end I am listing some specific issues and questions that should be considered and fixed in a new version of the article:

- p.2: "... in the spatial and temporal scales of nm to 10s of uM and us to 10s of seconds": Sorry, but it is impossible to understand what is meant here. Please rewrite this.

- p.3: "as an individual agent": this can be confusing, since many readers will think of agent-based simulations here. Perhaps it would be better to talk of invdividual particles / spheres.

- p.3: The text suggests that there is no volume exclusion for volume particles, but there is one on the membranes. Is this indeed so? If yes, why?

- p.3: "it is significantly faster": significantly faster can mean factor 1.2x faster, but it seems that the performance enhancement can be far higher in MCell4. I would therefore replace "significantly" by "dramatically" or similar, and name a concrete number / speed-up factor here.

- p.3: "And most of MCell's features ... have been retained.": Why "only" most, and not all? Was there technical problems, or were they deemed irrelevant?

- p.5/6: "molecule-wall collision": I would rather say "molecule-surface collision". Earlier "membranes" is used synonymously with "surfaces". It would be good to settle for one term here ("surfaces" seems most appropriate).

- p.6: What exactly is meant by "multi-physics" simulations? I only know this term from game software development, as in natural sciences simulations there should be only one physics (that arguably can be implemented in different ways, with varying degrees of accuracy).

- p.7: Why is the "DiffuseReactEvent" second in the schematic, but fourth in the caption? This is confusing. Please, on occasion, increase the arrow heads in all of these schematics.

- p.9: 3^10 seems an impressively high number, but it is actually "only" ~60.000... It would be good to write this out in order to avoid unjustified superlatives here.

- Fig. 6: What does "not all dependencies are shown" mean here? This seems strange...

- p. 12: The notation with comma/apostrophe for inwards/outwards is introduced twice and therefore repeating. Please check.

- p. 15: tilde notation for states ~0 and ~1: Why the choice of the tilde, and not simply state "0" and "1"?

- p. 17: Description of callback functions and reference links to code: This should not be part of the Results section, because it describes _Methods_. The results section here should be talking of the SNARE complex system and what we learn about it by using MCell4 simulations. Certainly, the results section of a PLoS Comp Biol article should not talk of particular Python classes that were implemented for obtaining some results. This purely technical content should part of a Methods section or the SI.

- p. 19: 17000 molecules / um^2: This appears a rather high molecule number. Is this the standard density usually simulated in MCell4? Then this should be mentioned. It remains a bit unclear, why for such a high density the surface is tiled into 40000 tiles / um^2 if only one molecule can occupy a tile at a time. Is it, in this setting, not highly probable that two molecules could end up in one tile?

- p. 21: "A demonstration that will be shown in this section ...": "Showing" and "demonstrating" are highly synonymous, so it seems that this introduction can be shortened. E.g. by saying "Here we show" or "Here we demonstrate this via ..."

- p. 22, Figs. 19 and 20: This can clearly be grouped into one figure. Please indicate what "protein R" is in the captions.<br 

---

## [Decision Letter · Decision Letter 1]

3 Jan 2024

Dear Dr. Bartol,

We are pleased to inform you that your manuscript 'MCELL4 WITH BIONETGEN A MONTE CARLO SIMULATOR OF RULE-BASED REACTION-DIFFUSION SYSTEMS WITH PYTHON INTERFACE' has been provisionally accepted for publication in PLOS Computational Biology.

Please have a look at the comments of reviewers 2 and 3 with important suggestions for final improvements that should be taken into account. 

Best regards,

Martin Meier-Schellersheim

Academic Editor

PLOS Computational Biology

Daniel Beard

Section Editor

PLOS Computational Biology

Reviewer's Responses to Questions

**Comments to the Authors:**

Reviewer #1: The authors have thoroughly addressed all of my concerns. The paper reads well, and the software represents an impressive advancement.

It does not need to be considered for further revisions, but I will reply to the comments made about the value of running benchmark tests (comment R1-12). Regarding the faster simulation of the model using MCell 3 in your test (34 s) when compared to mine (120 s), the difference undoubtedly arises from the faster computer. I ran the tests on a computer from about 2007, and you ran them on a 2017 computer. (I also ran them on a 2013 computer and measured 67 s (Andrews, 2018), which fits the trend.) Your comparison of MCell 4 vs. MCell 3 shows that the new code is 1.2 times faster, which is good news. By themselves, I agree that these benchmark numbers are fairly meaningless; however, when different software tools are compared on the same computer, then they give useful information about their relative speeds. I have worked hard to publish unbiased comparisons in my prior papers but would welcome the opportunity to collaborate on future comparisons.

Reviewed by Steve Andrews

Reviewer #2: uploaded as attachment

Reviewer #3: Overall I am very content with how the authors addressed the comments of me and the other reviewers and how they improved their manuscript.

I am particularly happy that the authors now explicitly mention the speed-up achieved in the new version of MCell, which in fact seems substantial and therefore constitutes an important improvement of the new software in addition to its increased user-friendliness and interconnectivity.

I would also like to thank them for the clarification on the format of the article, which indeed I was initially not aware of. In view of this clarification, their style of presentation makes much more sense.

In my opinion, this article seems good to go into publication now. I would only recommend the following minor additional changes, trusting that the authors would implement them in their own interest:

- I recommend enlarging Figures 5, 6 and 18 further, as to make the text labels therein more readable. In contrast to other figures of the paper, with these three figures enlargement still seems possible.

- On p.14, for clarity and consistency, it would be good to state explicitly that unit "N" refers to the particle number, as it is already done in the corresponding figure caption.

**Have the authors made all data and (if applicable) computational code underlying the findings in their manuscript fully available?**

Reviewer #1: Yes

Reviewer #2: Yes

Reviewer #3: Yes

PLOS authors have the option to publish the peer review history of their article (what does this mean?). If published, this will include your full peer review and any attached files.

Reviewer #1: **Yes: **Steven S. Andrews

Reviewer #2: No

Reviewer #3: No

---

## [Editor Report · Acceptance letter]

17 Apr 2024

PCOMPBIOL-D-23-00698R1 

MCELL4 WITH BIONETGEN A MONTE CARLO SIMULATOR OF RULE-BASED REACTION-DIFFUSION SYSTEMS WITH PYTHON INTERFACE

Dear Dr Bartol,

I am pleased to inform you that your manuscript has been formally accepted for publication in PLOS Computational Biology. Your manuscript is now with our production department and you will be notified of the publication date in due course.

With kind regards,

Judit Kozma
